# Research inefficiencies in external validation studies of the Framingham Wilson coronary heart disease risk rule: A systematic review

**Jong-Wook Ban**[1,2]*, **Lucy Abel**[3], **Richard Stevens**[3], **Rafael Perera**[3]

**1** Centre for Evidence-Based Medicine, University of Oxford, Oxford, United Kingdom, **2** Department for Continuing Education, University of Oxford, Oxford, United Kingdom, **3** Nuffield Department of Primary Care Health Sciences, University of Oxford, Oxford, United Kingdom

* drjwban@gmail.com

## Abstract

### Background

External validation studies create evidence about a clinical prediction rule's (CPR's) generalizability by evaluating and updating the CPR in populations different from those used in the derivation, and also by contributing to estimating its overall performance when meta-analysed in a systematic review. While most cardiovascular CPRs do not have any external validation, some CPRs have been externally validated repeatedly. Hence, we examined whether external validation studies of the Framingham Wilson coronary heart disease (CHD) risk rule contributed to generating evidence to their full potential.

### Methods

A forward citation search of the Framingham Wilson CHD risk rule's derivation study was conducted to identify studies that evaluated the Framingham Wilson CHD risk rule in different populations. For external validation studies of the Framingham Wilson CHD risk rule, we examined whether authors updated the Framingham Wilson CHD risk rule when it performed poorly. We also assessed the contribution of external validation studies to understanding the Predicted/Observed (P/O) event ratio and $c$ statistic of the Framingham Wilson CHD risk rule.

### Results

We identified 98 studies that evaluated the Framingham Wilson CHD risk rule; 40 of which were external validation studies. Of these 40 studies, 27 (67.5%) concluded the Framingham Wilson CHD risk rule performed poorly but did not update it. Of 23 external validation studies conducted with data that could be included in meta-analyses, 13 (56.5%) could not fully contribute to the meta-analyses of P/O ratio and/or $c$ statistic because these performance measures were neither reported nor could be calculated from provided data.

**Data Availability Statement:** All relevant data are within the manuscript and its Supporting information files.

**Funding:** The authors received no specific funding for this work.

**Competing interests:** Rafael Perera receives funding from the NIHR Oxford Biomedical Research Council (BRC), the NIHR Oxford Medtech and In-Vitro Diagnostics Co-operative (MIC), and the Oxford Martin School. Richard Stevens and Rafael Perera receive funding from the NIHR Applied Research Collaboration (ARC) Oxford and Thames Valley.

## Discussion

Most external validation studies failed to generate evidence about the Framingham Wilson CHD risk rule's generalizability to their full potential. Researchers might increase the value of external validation studies by presenting all relevant performance measures and by updating the CPR when it performs poorly.

## Introduction

External validation is a stage of clinical prediction rule (CPR) development where a CPR derived in a population is evaluated in different populations with a comparable health condition to the health condition of the derivation population (e.g., in another country) [1, 2]. An external validation study can generate evidence about a CPR's generalizability in two distinct ways [3].

Firstly, on a narrow scope, an external validation study can assess whether a CPR is applicable to a new population or setting [4, 5]. When a CPR performs adequately in an external validation study, clinicians can apply the CPR to populations and settings similar to those used in the external validation study, with confidence in its performance [6, 7]. However, CPRs often perform unsatisfactorily when applied to populations different from the ones they are derived, in external validation studies [8–10], which might be due to inflated accuracy from suboptimal design and methods used to derive the CPRs or differences in populations and settings between derivation and external validation studies [4, 11].

Sometimes, authors of external validation studies declare that the CPRs, when they perform poorly, are not generalizable [12–14]. There are several disadvantages for this approach. When no valid CPR exists for the population, clinicians cannot make decisions guided by a CPR until a new accurate CPR is created by often repeating the entire derivation process from ground up, because using poorly performing CPR can lead to decisions harmful to patients [15]. Moreover, this means that the knowledge accumulated while deriving the CPR is simply wasted [11, 16]. Further, this approach might promote the proliferation of many redundant CPRs for the same problem [11, 16] because the poor performance of CPR is one of the most commonly stated justifications by authors who derive a new cardiovascular CPR [17].

Alternatively, authors could attempt to update the CPR while conducting the external validation study when it performs poorly [3, 11, 16, 18]. The performance of the CPR could improve simply by adjusting the calibration intercept or by re-estimating coefficients of predictor variables [11, 19–21]. To the contrary to dismissing the CPR that performs poorly in external validation, previous research and evidence accrued is efficiently used when the CPR is updated [11, 16, 22]. Although the updated CPR still should be externally validated, with this approach, it might be unnecessary for researchers to create a new CPR just for this population by rerunning the entire derivation process [11, 16]. For example, the REGICO function was created by calibrating the Framingham Wilcon CHD risk rule in Spanish cohort [23], which has been successfully validated in various Spanish cohorts [24–26], and adopted by Spanish clinical practice guidelines [27, 28].

Secondly, and more broadly, the results of external validation studies of a CPR can contribute to understanding the CPR's overall performance across an overarching population with the same health condition when meta-analysed in a systematic review [3]. By understanding the overall performance of a CPR from the meta-analysis results, clinicians can guide their use of the CPR in practice. For example, a systematic review and meta-analysis of external

validation studies showed that the American College of Cardiology and American Heart Association (ACC/AHA) pooled cohort equation frequently overestimates cardiovascular disease (CVD) risk [21]. Based on these results, clinicians could use caution when interpreting a person's CVD risk estimated with the ACC/AHA pooled cohort equation.

For the results of an external validation study to contribute to a meta-analysis, relevant performance measures of a CPR must be provided. According to the TRIPOD statement published in 2015, calibration and discrimination measures "should be reported in all prediction model papers" [29]. However, systematic reviews have shown that external validation studies often do not report recommended performance measures, especially measures of calibration [8, 30, 31]. Most studies included in these systematic reviews predate the TRIPOD statement, and authors might not have guidance for appropriate reporting. However, it is still true that when external validation studies do not report relevant performance measures, they might not contribute to understanding a CPR's overall performance in a meta-analysis to their full potential, limiting the value of external validation studies.

While most cardiovascular CPRs do not have a timely conducted independent external validation study [32], there are a small number of cardiovascular CPRs for which many external validation studies have been performed repeatedly. In 1998, Wilson et al. [33] derived a CPR for coronary heart disease (CHD) risk using the Framingham Heart Study cohort. The Framingham Wilson CHD risk rule estimates the 10-year risk of CHD (composite of angina pectoris, acute myocardial infarction, coronary insufficiency, and CHD death) and hard CHD (all CHD outcomes except for angina pectoris) with the following predictors: age, cigarette use, diabetes mellitus status, total and high-density lipoprotein cholesterol categories, and blood pressure categories [33]. Systematic reviews [31, 34] have shown that more external validation studies for the Framingham Wilson CHD risk rule have been conducted than any other cardiovascular CPR. For example, a systematic review found that the Framingham Wilson CHD risk rule was the most frequently validated cardiovascular CPR with 89 external validation studies [31].

Therefore, in this study, we aimed to explore research inefficiencies of external validation studies of the Framingham Wilson CHD risk rule; firstly by assessing whether authors of external validation studies updated the Framingham Wilson CHD risk rule when it performed poorly, and then by evaluating whether external validation studies contributed to understanding the overall performance of the Framingham Wilson CHD risk rule.

## Materials and methods

### Information source and search for external validation studies

In November 2015, a forward citation search of the derivation study of the Framingham Wilson CHD risk rule [33] was conducted in Scopus. This initial search was updated in June 2020 and December 2022. We conducted our forward citation search using Scopus as a citation index because we have previously demonstrated that this method could efficiently and systematically identify CPR studies [32, 35]. Further, for citation searches in the health science field, Scopus tends to produce more robust results than Web of Science while minimally missing unique references [36, 37]. The searches were not limited to a specific language, date range, access type, or document type.

### Study selection

The title and abstract of all identified references were screened, and the full-text article of relevant references was reviewed. A study was included, if

1. the CPR evaluated in the study could be unambiguously identified as the Framingham Wilson CHD risk rule,

2. the Framingham Wilson CHD risk rule was applied to a new population different from the one used in the derivation study,

3. the cardiovascular outcome evaluated was either CHD (a composite of angina pectoris, acute myocardial infarction, coronary insufficiency, and CHD death) or hard CHD (a composite of acute myocardial infarction, coronary insufficiency, and CHD death) with an average follow-up of five years or longer, and

4. a valid performance measure of a CPR such as discrimination or calibration was reported or could be estimated using data provided in the study.

To be unambiguously identified as a study evaluating Framingham Wilson CHD risk rule, authors of the study must have identified the CPR using the term "Framingham" or "Wilson" with a citation to the derivation study of the Framingham Wilson CHD risk rule [33] or presented sufficient details about the CPR including baseline survival function, regression coefficient, hazard ratio, and scoring systems [38] for the verification. When studies defined CHD outcomes using criteria different from the ones used by the Framingham Heart Study [39] due to evolving practice standards over time, we judged whether the modifications were relevant to current clinical practice. We included studies with an average follow-up of five years or longer because, for the vast majority of CPRs for cardiovascular disease risk, this was the intended timeframe for predicting outcomes according to a systematic review [31]. Studies that evaluated modified versions of the Framingham Wilson CHD risk rule were excluded.

## Data collection

From each study meeting the inclusion criteria, the following data were collected: bibliographic information, cohort, geographic location, number of participants and events, duration of follow-up, and performance measures. The included studies were categorised into two groups: (1) studies in which one of the stated aims was externally validating Framingham Wilson CHD risk rule, which will be referred to as "external validation studies" in our study, and (2) studies that did not aim to externally validate the Framingham Wilson CHD risk rule but in which a valid performance measure of a CPR was reported or could be calculated using provided data.

For studies whose stated aims included externally validating the Framingham Wilson CHD risk rule, we assessed whether authors concluded that the Framingham Wilson CHD risk rule performed adequately or inadequately. Authors' conclusions were interpreted based on their use of words such as "underestimate", "overestimate", "well", "poorly", "adequate", "inadequate", and other similar terms in the abstract, results, and discussion sections. For external validation studies where authors concluded the Framingham Wilson CHD risk rule performed inadequately, we also examined whether the authors updated the Framingham Wilson CHD risk rule.

We also included studies in which a performance measure of the Framingham Wilson CHD risk rule could be obtained even if the stated aim was not external validation of the Framingham Wilson CHD risk rule, because these studies could contribute to meta-analysing the performance measures of the Framingham Wilson CHD risk rule. The first author (JWB) conducted the forward citation searches, screened the references found, reviewed the full-text articles, assessed the eligibility, and collected data from included studies. A second reviewer (LA)

verified the eligibility of included articles and collected all data independently. Any disagreements were adjudicated through discussions.

## Outcome measures

Research inefficiency occurs when research studies fail to "contribute to knowledge or to practice and policy," to their maximum potential [22, 40]. In our study, we assessed two types of inefficiency among external validation studies of the Framingham Wilson CHD risk rule. Firstly, we evaluated the inefficiency from external validation studies that do not attempt to update the Framingham Wilson CHD risk rule when it performs poorly; thus, missing the opportunity to create a CHD risk rule that could be useful for populations similar to the one used in the external validation. Therefore, one of our principal outcome measures was the proportion of external validation studies that concluded the Framingham Wilson CHD risk rule performed inadequately but did not update the CPR.

Secondly, we assessed the inefficiency from external validation studies that are potentially eligible to be included in a meta-analysis and contribute to understanding the performance of the Framingham Wilson CHD risk rule, but fail to do so because either a P/O ratio or $c$ statistic is not reported or cannot be obtained from provided data. Thus, our other main outcome measure was the proportion of eligible external validation studies that could not be included in the meta-analysis of P/O ratio or $c$ statistic due to the insufficient reporting of relevant performance measures. We chose to evaluate the P/O ratio because existing systematic reviews [21, 41, 42] of the Framingham Wilson CHD risk rule invariably meta-analyzed the P/O ratio to summarize calibration.

## Data analysis

Using all included external validation studies, we calculated the proportion of studies in which authors concluded that the Framingham Wilson CHD risk rule (a) performed adequately, (b) performed inadequately and updated the CPR, and (c) performed inadequately but did not update the CPR.

Arguably, a meta-analysis of external validation studies should include studies conducted in populations that are "plausibly related" [43], which is not always easy to determine clearly. In our study, we took a conservative approach and included in meta-analyses only the studies conducted in the USA and European geographic regions with comparable cardiovascular disease burden, the prevalence of risk factors, and healthcare environment [44]: the UK, Northern Europe, Western Europe, and Southern Europe. We meta-analysed the Predicted/Observed (P/O) event ratio and $c$ statistic of the Framingham Wilson CHD risk rule. The definitions of CHD from external validation studies included in the meta-analyses are summarized in S1 Table. Random effects meta-analyses were conducted using methods described by Debray et al. [45], as described in S1 Appendix. When multiple studies were conducted using data from the same cohort, only the study with the largest dataset was included in the meta-analyses. At the suggestion of a reviewer, we display results in forest plots without summary estimates when heterogeneity is high, since summary estimates from meta-analyses are incidental to our study aims rather than directly relevant to our research questions. We presented the proportion of eligible external validation studies that could not be included in the meta-analysis of P/O ratio or $c$ statistic because a relevant performance measure was not reported or could not be estimated from provided data.

Additionally, we conducted post hoc citation analyses to evaluate the potential impact of external validation studies in which authors concluded that the Framingham Wilson CHD risk rule performed inadequately but did not update it. In Scopus, we ran forward citation

searches of these external validation studies, which were updated in December 2022. We searched for studies that derived a new cardiovascular CPR while using the Framingham Wilson CHD risk rule's poor performance as one of the justifications and citing one of these external validation studies.

We prepared this report following all applicable recommendations from the Preferred Reporting Items for Systematic Reviews and Meta-Analyses (PRISMA) statement [46] as summarised in S2 Appendix. Data from our study are provided in S1 Dataset.

## Results

### Included studies of the Framingham Wilson CHD risk rule

A total of 98 studies that evaluated the Framingham Wilson CHD risk rule from 84 publications were included, Fig 1.

Characteristics of the included studies are summarised in Table 1. For 40 (40.8%) of 98 included studies, one of the aims was externally validating the Framingham Wilson CHD risk rule. For 58 (59.2%) studies, although aims did not include externally validating the Framingham Wilson CHD risk rule, a performance measure of the Framingham Wilson CHD risk rule was reported or could be estimated using data provided in the study. The lists of included articles, excluded articles, and journals that published the included studies are provided in S3 Appendix, S2 and S3 Tables, respectively.

Performance measures of the Framingham Wilson CHD risk rule reported by the included studies are summarised in Table 2. Of 40 studies that aimed to externally validate the Framingham Wilson CHD risk rule, 17 (42.5%) did not report any calibration measure. Only 14 (35.0%) studies reported calibration according to the recommendations from the TRIPOD statement: 5 (12.5%) studies with both calibration table and plot, 8 (20.0%) studies with a calibration plot only, one (2.5%) study with a calibration table only. A measure of discrimination was not reported in a minority (11 of 40, 27.5%) of studies that aimed to externally validate the Framingham Wilson CHD risk rule. However, $c$ statistic was reported with a 95% confidence interval only in 17 of 40 (42.5%) external validation studies, adhering to the recommendation in the TRIPOD statement.

### Inefficiency from failure to update the Framingham Wilson CHD risk rule

Authors' interpretations of the Framingham Wilson CHD risk rule's performance in external validation study and their consequent actions are summarised in Fig 2. Of 40 external validation studies of the Framingham Wilson CHD risk rule, authors of 27 external validation studies (67.5%) concluded that the performance was inadequate but did not update the CPR. On the other hand, the authors of four external validation studies (10.0%) that concluded the Framingham Wilson CHD risk rule performed poorly updated the CPR: two studies by recalibrating [47, 48] and two studies by re-estimating parameters and modifying predictor variables [49, 50]. In three of these four studies, the performance of the Framingham Wilson CHD risk rule was judged to be improved after updating it.

In a post hoc citation analysis, a total of 1,341 references were found in forward citation searches of 27 external validation studies that concluded the performance of the Framingham Wilson CHD risk rule was inadequate but did not update it. From these references, we found 20 studies that derived a new cardiovascular CPR using the poor performance of the Framingham Wilson CHD risk rule in the external validation study as one of the justifications, with a citation to the external validation study. Of these 20 new cardiovascular CPRs, four were derived by overlapping authors with external validation studies, and 16 were derived by different authors, as presented in S4 Table.

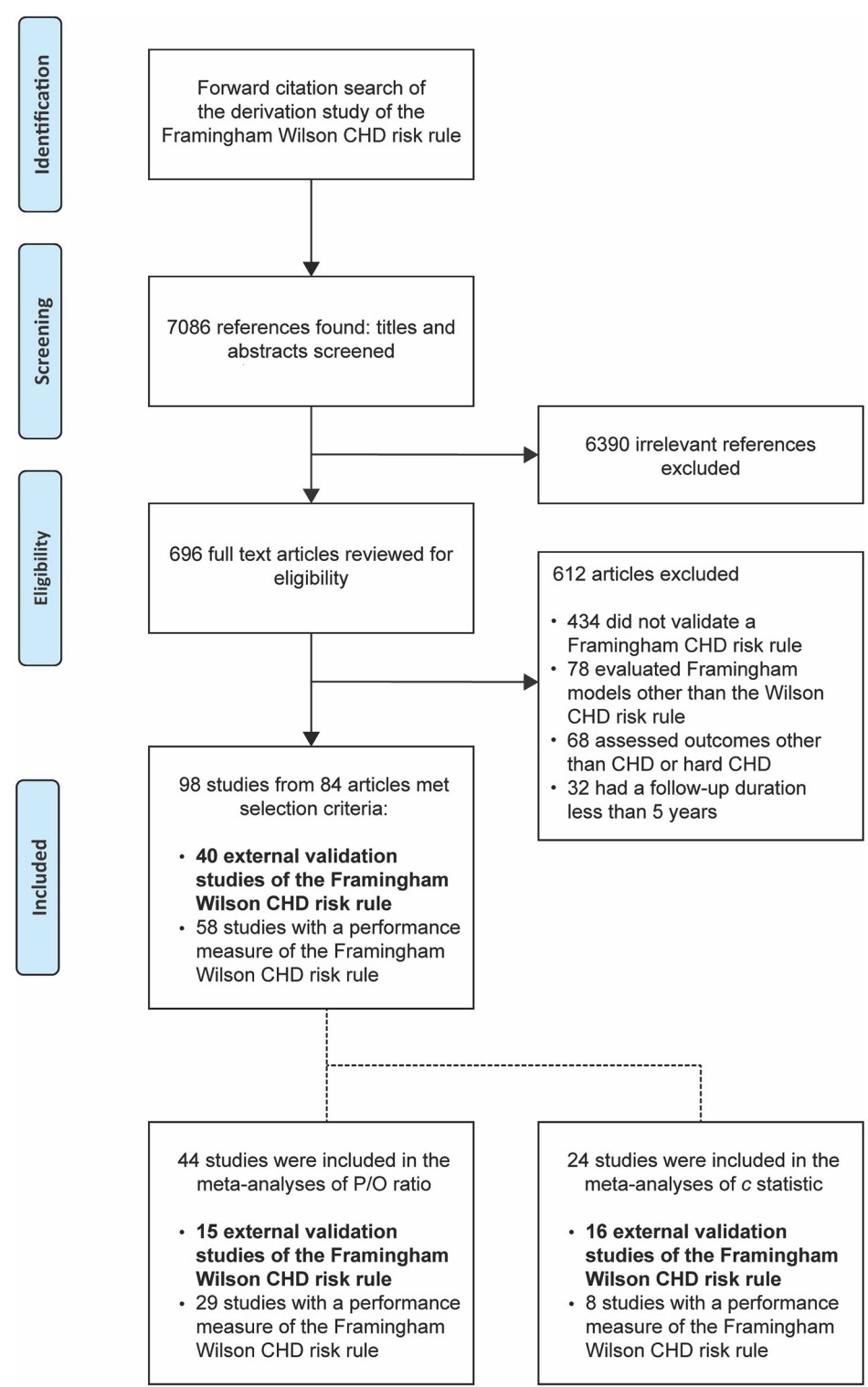

**Fig 1. Search and selection of studies that evaluated the Framingham Wilson coronary heart disease (CHD) risk rule.** P/O ratio: Predicted/Observed event ratio.

**Table 1. Characteristics of the included studies of the Framingham Wilson coronary heart disease (CHD) risk rule.**

| Characteristic | External validation studies of the Framingham Wilson CHD risk rule, n = 40 | Other studies with a performance measure of the Framingham Wilson CHD risk rule, n = 58 | All included studies, n = 98 |
|---|---|---|---|
| Publication year, median (IQR) | 2008 (2006–2012) | 2010 (2008–2014) | 2009 (2007–2014) |
| *Journal impact factor, median (IQR) | 4.213 (3.133–8.754) | 13.2825 (5.099–51.598) | 7.05 (3.35–35.855) |
| Study location | | | |
| • Within the USA | 7 (17.5) | 21 (36.2) | 28 (28.6) |
| • Outside the USA | 33 (82.5) | 37 (63.8) | 70 (71.4) |
| Sample size, median (IQR) | 1,437.5 (625.5–5,611) | 3,124.5 (1,392–6,604) | 2,474.5 (847–5,732) |
| Number of events, median (IQR) | 131.5 (62–294) | 204 (117–499) | 187 (93–460) |
| Duration of follow-up in years, median (IQR) | 10.0 (7.65–10.0) | 8.6 (6.4–10.9) | 9.65 (6.4–10.0) |
| Description of CPR being validated | | | |
| • Risk equation [a] | 7 (17.5) | 17 (29.3) | 24 (24.5) |
| • Risk score [b] | 4 (10.0) | 9 (15.5) | 13 (13.3) |
| • Unclear | 29 (72.5) | 32 (55.2) | 61 (62.2) |
| Handling of missing data | | | |
| • Complete case analysis | 30 (75.0) | 27 (46.6) | 57 (58.2) |
| • Imputation | 0 (0.0) | 16 (27.6) | 16 (16.3) |
| • Unclear | 10 (25.0) | 15 (25.9) | 25 (25.5) |

Values are numbers (%) unless indicated otherwise.

* 2021 Web of Science Journal Citation Report.

[a] Risk equation: regression equations that estimate the 10-year risk of developing CHD using coefficients of predictor variables.

[b] Risk score: simplified scoring systems that categorise the 10-year risk of developing CHD based on the sum of points assigned to applicable predictor variables.

## Inefficiency from the inability to contribute to understanding the overall performance of the Framingham Wilson CHD risk rule

A total of 44 studies were included in the meta-analysis of the P/O ratio: 15 studies with an explicit aim of externally validating the Framingham Wilson CHD risk rule and 29 studies where a P/O ratio could be obtained from data. As presented in Fig 3A and 3B, the P/O ratios that were obtained showed high heterogeneity, with $I^2$ statistic greater than 80% in all analyses.

A total of 24 studies were included in the meta-analysis of $c$ statistic: 16 studies with an explicit aim of externally validating the Framingham Wilson CHD risk rule and 8 studies where a $c$ statistic could be obtained from data. Forest plots of the $c$ statistic of the Framingham Wilson CHD risk rule are presented in Fig 4. Studies conducted in the USA, Northern Europe, and Southern Europe had high heterogeneity with a corresponding $I^2$ statistic of 96.98%, 96.50%, and 79.06%, respectively. The summary $c$ statistic (95% confidence interval) from the meta-analysis of three studies conducted in the UK was 0.699 (0.680–0.718) with an $I^2$ statistic of 30.33%. The summary $c$ statistic (95% confidence intervals) from the meta-analyses of five Western European studies was 0.692 (0.660–0.722) with an $I^2$ statistic of 49.00%.

Of 40 studies that aimed to externally validate the Framingham Wilson CHD risk rule, 23 studies were conducted in the USA and Europe: therefore could have been included in the meta-analyses as presented in Table 3. All 23 studies were eligible to be included in the meta-analysis of $c$ static. However, only 21 of 23 studies were eligible for the meta-analysis of the P/O ratio; two studies, Vaidya 2007 [51] and Simmons 2008 [52], were ineligible because they were conducted using a subset of data from the same cohort used by another eligible study. Six

**Table 2. Performance measures of the Framingham Wilson coronary heart disease (CHD) risk rule reported by included studies.**

| Performance measure | External validation studies of the Framingham Wilson CHD risk rule, n = 40 | Other studies with a performance measure of the Framingham Wilson CHD risk rule, n = 58 | All included studies, n = 98 |
|---|---|---|---|
| Any calibration | 23 (57.5) | 9 (15.5) | 32 (32.7) |
| • P/O ratio with 95% CI | 3 (7.5) | 0 (0.0) | 3 (3.1) |
| • P/O ratio without 95% CI | 5 (12.5) | 0 (0.0) | 5 (5.1) |
| • Calibration plot | 13 (32.5) | 1 (1.7) | 14 (14.3) |
| • Calibration table | 6 (15.0) | 0 (0.0) | 6 (6.1) |
| • Calibration intercept | 2 (5.0) | 0 (0.0) | 2 (2.0) |
| • Calibration slope | 1 (2.5) | 1 (1.7) | 2 (2.0) |
| • Goodness of fit test | 9 (22.5) | 7 (12.1) | 16 (16.3) |
| Any discrimination | 29 (72.5) | 21 (36.2) | 50 (51.0) |
| • $c$ statistic with 95% CI | 17 (42.5) | 9 (15.5) | 26 (26.5) |
| • $c$ statistic without 95% CI | 11 (27.5) | 12 (20.7) | 23 (23.5) |
| • ROC curve | 16 (40.0) | 5 (8.6) | 21 (21.4) |
| • Discrimination slope | 1 (2.5) | 1 (1.7) | 2 (2.0) |
| Any classification | 9 (22.5) | 0 (0.0) | 9 (9.2) |
| • Sensitivity or specificity | 9 (22.5) | 0 (0.0) | 9 (9.2) |
| • Positive or negative predictive value | 8 (20.0) | 0 (0.0) | 8 (8.2) |
| • Positive or negative likelihood ratio | 5 (12.5) | 0 (0.0) | 5 (5.1) |
| • Diagnostic odds ratio | 4 (10.0) | 0 (0.0) | 4 (4.1) |
| • Accuracy | 4 (10.0) | 0 (0.0) | 4 (4.1) |
| Other* | 1 (2.5) | 3 (5.2) | 4 (4.1) |

Values are the number of studies (%).

ROC: receiver operating characteristic.

* Includes measures such as $R^2$ and Brier score.

external validation studies of the Framingham Wilson CHD risk rule could not be included the meta-analysis of P/O ratio because they neither reported a P/O ratio nor provided sufficient information about the data to estimate it. Similarly, seven external validation studies of the Framingham Wilson CHD risk rule could not contribute to the meta-analysis of $c$ statistic because they neither reported a $c$ statistic nor provided data to estimate it.

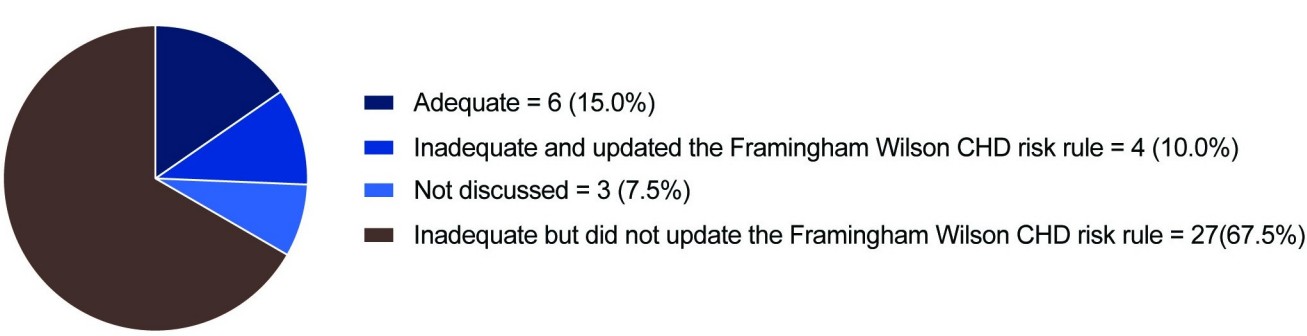

- Adequate = 6 (15.0%)
- Inadequate and updated the Framingham Wilson CHD risk rule = 4 (10.0%)
- Not discussed = 3 (7.5%)
- Inadequate but did not update the Framingham Wilson CHD risk rule = 27 (67.5%)

Total = 40

**Fig 2. Authors' interpretations of the Framingham Wilson coronary heart disease (CHD) risk rule's performance in external validation study and consequent actions.**

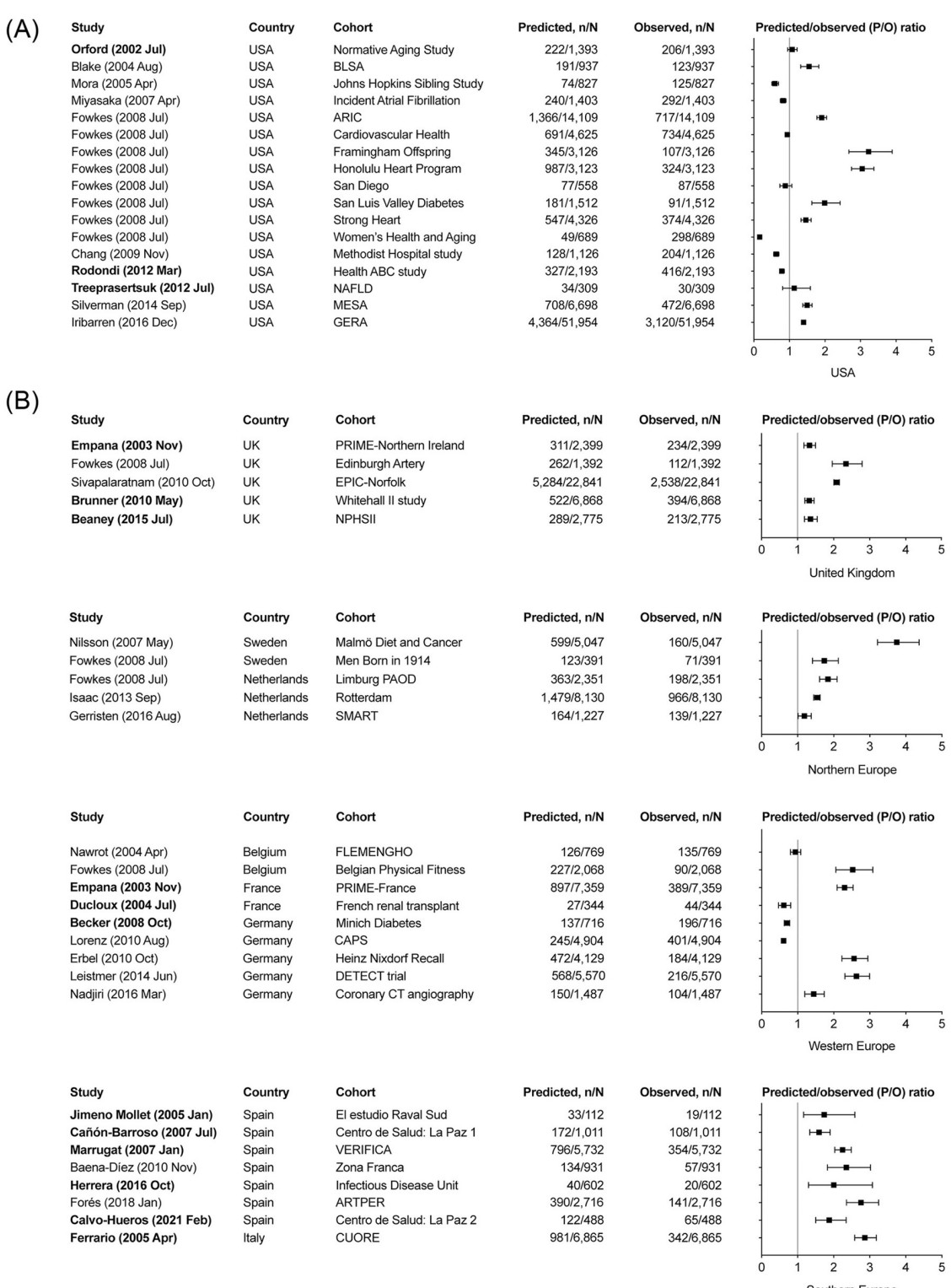

**Fig 3. A.** Meta-analyses of Predicted/Observed (P/O) ratio from studies conducted in the USA. Studies with an explicit aim to externally validate the Framingham Wilson coronary heart disease rule are presented in bold characters. n: the number of participants with the outcome, N: the total number of participants. **B.** Meta-analyses of Predicted/Observed (P/O) ratio from studies conducted in Europe. Studies with an explicit aim to externally validate the Framingham Wilson coronary heart disease rule are presented in bold characters. n: the number of participants with the outcome, N: the total number of participants.

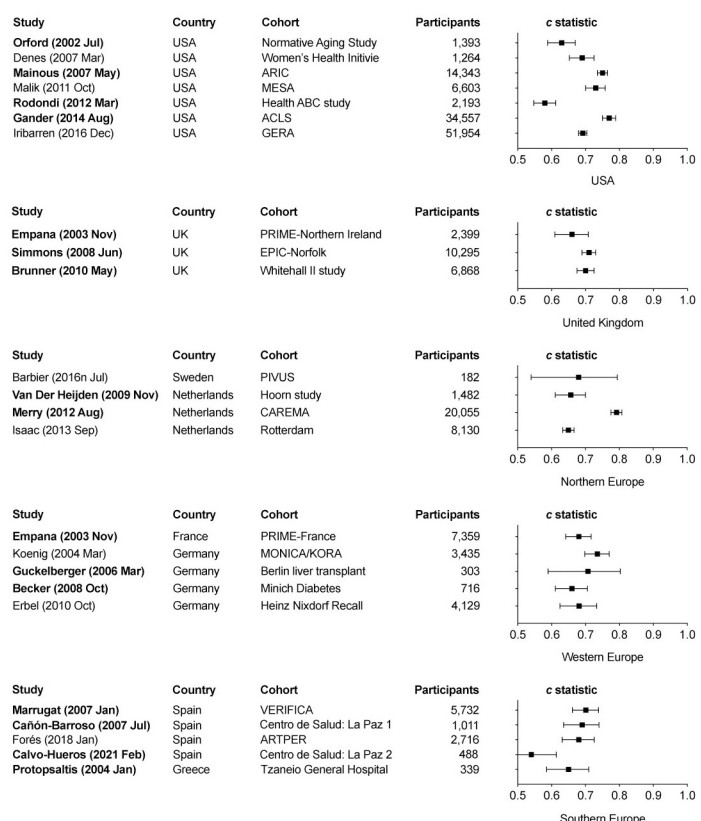

**Fig 4. Meta-analyses of *c* statistic from studies conducted in the United States and Europe.** Studies with an explicit aim to externally validate the Framingham Wilson coronary heart disease rule are presented in bold characters.

In summary, of 23 studies that aimed to externally validate the Framingham Wilson CHD risk rule, only ten (43.5%) contributed to understanding its performance in eligible meta-analyses of P/O ratio and *c* statistic. On the other hand, 13 (56.5%) studies that aimed to externally validate the Framingham Wilson CHD risk rule did not contribute to eligible meta-analyses to their full potential because they neither reported relevant performance measures nor provided data to estimate them.

## Discussion

In this study, we evaluated whether external validation studies of the Framingham Wilson CHD risk rule generated useful evidence for new populations or settings, and whether they contributed to understanding the Framingham Wilson CHD risk rule's overall performance in overarching populations.

### Summary of findings

Many (67.5%) external validation studies found that the Framingham Wilson CHD risk rule performed inadequately but did not take the opportunity to update the CPR, which we consider an inefficient use of research and data. Subsequently, 20 new cardiovascular CPRs were derived using the poor performance of the Framingham Wilson CHD risk rule in these external validation studies as one of the justifications.

**Table 3. Performance measures reported or calculated from data provided in 23 external validation studies of the Framingham Wilson coronary heart disease risk rule, conducted in the USA and Europe.**

| Study | Location | Cohort | Performance measure reported or calculated from data | |
|---|---|---|---|---|
| | | | P/O ratio | c statistic |
| Orford 2002 [53] | USA | Normative Aging Study | Calculated from data | Reported |
| Mainous 2007 [54] | USA | ARIC | Not available* | Reported[b] |
| Vaidya 2007 [51] | USA | Johns Hopkins Sibling study | Ineligible[a] | Not available* |
| Rodondi 2012 [49] | USA | Health ABC study | Calculated from data | Reported |
| Treeprasertsuk 2012 [55] | USA | NAFLD | Calculated from data | Not available* |
| Gander 2014 [56] | USA | ACLS | Not available* | Reported[b] |
| Emapana 2003 [57] | UK | PRIME-UK | Reported[b] | Reported |
| Simmons 2008 [52] | UK | EPIC-Norfolk | Ineligible[a] | Reported[b] |
| Brunner 2010 [58] | UK | Whitehall II study | Calculated from data | Reported[b] |
| Beaney 2015 [59] | UK | NPHSII | Calculated from data | Not available* |
| Van Der Heijden 2009 [14] | Northern Europe | Hoorn study | Not available* | Reported[b] |
| Merry 2012 [60] | Northern Europe | CAREMA | Not available* | Reported[b] |
| Empana 2003 [61] | Western Europe | PRIME- France | Reported[b] | Reported |
| Ducloux 2004 [62] | Western Europe | French renal transplant cohort | Calculated from data | Not available* |
| Guckelberger 2006 [63] | Western Europe | Charité-Universitätsmedizin Berlin liver transplant | Not available* | Reported |
| Becker 2008 [64] | Western Europe | Munich Diabetes | Calculated from data | Reported[b] |
| Jimeno-Mollet 2005 [65] | Southern Europe | El estudio Raval Sud | Calculated from data | Not available* |
| Ferrario 2005 [47] | Southern Europe | CUORE | Calculated from data | Not available* |
| Marrugat 2007 [24] | Southern Europe | VERIFICA | Calculated from data | Reported |
| Cañón-Barroso 2007 [66] | Southern Europe | Centro de Salud: La Paz 1 | Calculated from data | Reported[b] |
| Herrera 2016 [26] | Southern Europe | Infectious Disease Unit at Hospital del Mar | Calculated from data | Not available* |
| Calvo-Hueros [67] | Southern Europe | Centro de Salud: La Paz 2 | Reported | Reported[b] |
| Protopsaltis 2004 [68] | Southern Europe | Tzaneio General Hospital | Not available* | Reported |

P/O ratio: Predicted/Observed event ratio.

* Not reported by the authors of external validation studies and could not be calculated from data provided in the external validation studies despite being conducted using data that could contribute to the meta-analyses.

[a] These studies were ineligible because they were conducted using a subset of data from the same cohort used by another eligible study.

[b] Reported with a 95% confidence interval.

Many external validation studies of the Framingham Wilson CHD risk rule did not take the opportunity to report relevant calibration and discrimination measures. Even when authors of the external validation studies reported these performance measures of the Framingham Wilson CHD risk rule, most did not adhere to the recommendations from the TRIPOD statement. Consequently, the majority (56.5%) of external validation studies of the Framingham Wilson CHD risk rule fail to contribute to understanding the CPR's overall performance to full potential because they neither reported relevant performance measures nor provided sufficient information about the data to estimate them. These research waste and inefficiencies limit the value of external validation studies.

The heterogeneity between studies suggests that even when many previous validations have been conducted in a particular geographic location, there is still potential to learn something new due to differences in populations, changes in predictor and outcome definitions, and temporal evolution of clinical practice [69]. In particular, since the Framingham Heart Study defined these outcomes several decades ago, standard practices for diagnosing CHD have

evolved greatly [70–72]. Therefore, we do not argue that the validation studies were wasteful in themselves. Rather, we encourage authors of validation studies to maximise the value of their study: by publishing comprehensive validation statistics and by publishing an updated ("recalibrated") rule when indicated.

## Comparison with existing literature

Numerous CPRs for cardiovascular conditions have been derived [31, 34, 73, 74]. But, for some clinical conditions such as coronary heart disease or cerebrovascular accident, researchers keep deriving new CPRs despite many existing CPRs for the conditions. For instance, 363 distinct CPRs for assessing the risk of cardiovascular disease general population were found in a systematic review [31]. We found that, when authors of external validation studies concluded that the Framingham Wilson CHD risk rule performed poorly but did not attempt to update it, their conclusions were used as a rationale for deriving multiple, potentially redundant cardiovascular CPRs. These results are compatible with findings from a previous study that showed one of the most common justifications for deriving a new cardiovascular CPR was the poor performance in external validation [17]. Further, these new cardiovascular CPRs, added to the pool of hundreds of existing cardiovascular CPRs, are unlikely to be recognized or used by clinicians because most clinicians are familiar with and use only a limited number of cardiovascular CPRs [75, 76].

Many systematic reviews have pointed out various methodological and reporting issues in external validation studies. For example, systematic reviews have shown that many external validation studies include an insufficient number of participants with and without outcome events [8, 77], which can result in biased and imprecise estimation of predictive performance measures [78, 79]. Despite this, external validation studies infrequently describe how the sample size is justified [80, 81] and missing data in external validation studies are often handled by complete case analysis, which is deprecated [8, 77, 82]. Also, we found that many external validation studies of the Framingham Wilson CHD risk rule did not report calibration and discrimination measures as recommended by the TRIPOD statement. These findings are consistent with systematic reviews in diabetes, cancer, chronic kidney disease, and cardiovascular disease that showed external validation studies often do not report recommended performance measures, especially for calibration [8, 30, 31, 83, 84]. For example, Collins et al. [83] systematically reviewed studies externally validating CPRs from various clinical domains, and found that the majority (67.9%) of external validation studies did not report any calibration measures while a measure of discrimination was reported in most studies (73.1%). In addition, we found that performance measures of the Framingham Wilson CHD risk rule are often presented without their confidence intervals, although the TRIPOD statement recommends "reporting measures with confidence intervals" [38]. This is also in line with findings from existing systematic reviews [8, 84, 85]. Other systematic reviews have also shown that, while timely conducted independent external validation studies are scarce [9, 32], efforts to externally validate CPRs are disproportionately focused on a small number of CPRs such as the Framingham risk scores [10, 31, 34].

## Strengths and limitations

We included 98 studies from 84 publications (40 external validation studies and 58 studies with a performance measure of the Framingham Wilson CHD risk rule) that could potentially contribute to meta-analyses of Framingham Wilson CHD risk rule by conducting a forward citation search, which is considerably more than those included in a similar systematic review [21] that used exhaustive search strategy. This re-demonstrates the efficiency and robustness of the forward citation search used in this study as a method to identify external validation studies.

On the other hand, we only relied on Scopus to conduct our forward citation search. Although the use of a single citation index is a very common practice [86], and Scopus performs well compared with Web of Science in the health science field [36, 37], searching multiple sources might have been more ideal to ensure none of the potentially relevant references was missed.

Our study has the following limitations. We used the Framingham Wilson CHD risk rule as an example and it is uncertain that findings are generalisable to other CPRs that have many external validation studies. In addition, our investigation into studies that cite studies in our review (those that conclude Framingham is inadequate, but do not update) was a post hoc addition; after discovering that most external validation studies simply dismissed the Framingham Wilson CHD risk rule when it performed poorly, we considered it necessary to understand the impact of these studies.

We assessed whether external validation studies of the Framingham Wilson CHD risk rule reported its performace measure following the recommendations from the TRIPOD statement, which was published in 2015. Because most external validation studies (all except for three) that we evaluated predate the publication of the TRIPOD statement, our findings should not be interpreted as an assessment of reporting quality against the recommendations from the TRIPOD statement. Our intention was to investigate whether external validation studies provide sufficient information so that their results could contribute to the understanding the overall performance of the Framingham Wilson CHD risk rule.

We only considered overall calibration (P/O ratio) and discrimination ($c$ statistic) in particular; good validation should consider three or more dimensions [87, 88]. However, focussing on two commonly reported aspects of validation did not prevent us from demonstrating research inefficiencies in this literature. Also, we followed the approaches of Debray et al. [45], as described in S1 Appendix, for extracting P/O ratios; arguably, a systematic reviewer with sufficient software and mathematical knowledge could also estimate P/O ratios from published calibration plots, albeit inexactly; this would increase the number of external validation studies of the Framingham Wilson CHD risk rule that contributed to understanding the performance in eligible meta-analyses of P/O ratio and $c$ statistic from 10 of 23 (43.5%) to 11 (47.8%), but not change our overall conclusions.

Lastly, some of the external validation studies that we included were conducted using data from the same cohorts. For example, nine external validation studies were produced using data from Centro de Salud: La Paz [12, 66, 89–93], which had a total sample size of 1,011 with only 108 CHD events. Although it is arguably wasteful to repeatedly publish external validation studies of a prediction rule using a small dataset, we did not attempt to quantify whether these studies constitute research inefficiency or waste in our study.

## Research and clinical implications

In this study, we identified two key inefficiencies in the external validation step of cardiovascular CPR development. Future studies (e.g. qualitative interviews or analysis of CPR registry) could further explore causes and mechanisms for these inefficiencies. Understanding factors that influence these inefficiencies to occur might help develop targeted ways to prevent them and increase the chance for external validation studies to fully contribute to knowledge and practice.

Researchers planning to externally validate a CPR should systematically evaluate existing evidence and determine whether a new external validation study is needed. When the CPR performs well, clinicians are more likely to have confidence in its prediction and adopt the CPR in practice [94, 95]. But, when the CPR performs less than satisfactorily in an external validation study, researchers should avoid simply dismissing the CPR. Instead, researchers should attempt to update the CPR and evaluate whether a clinically meaningful predictive performance can be

achieved so that the evidence generated is useful to the relevant populations and settings. Journal editors should encourage authors to provide a clear justification for conducting an external validation study of a CPR, ideally based on a systematic examination of existing evidence, and to update the CPR when it performs poorly. Further, journals should adopt the TRIPOD statement in their instructions to authors and require authors to present all relevant performance measures in external validation studies so that other researchers and clinicians could use the results of external validation studies to further evaluate and implement the CPR.

Conducting a systematic review or using evidence from systematic reviews to determine whether a new external validation study is justified can be challenging [96, 97]. Creating a living evidence system for a CPR that continuously updates meta-analyses with the latest evidence of generalizability about the CPR [98] might be useful for researchers considering an external validation study as well as clinicians and other stakeholders of CPR research. Clinicians should be made aware that a CPR that performed poorly in an external validation study might have an updated version that performs well in similar populations [24, 26, 99].

## Conclusion

The focus of cardiovascular CPR research should shift from deriving many new CPRs to externally validating existing CPRs and updating them when needed because the potential end-users, such as patients, clinicians, researchers, and decision-makers, need to understand the generalizability of the CPRs. However, we have shown that conducting more external validation studies of a CPR does not always strengthen the CPR's evidence of generalizability. Further, we demonstrated that dismissing a CPR when it performs poorly can lead to the creation of new, potentially redundant CPRs. Therefore, in addition to shifting the focus of CPR research, the authors of external validation studies should ensure that they are using appropriate design and methods and providing sufficient information about the results so that the evidence they generate can help the end-users about the generalizability of the CPRs.

## Supporting information

**S1 Table. The definition of coronary heart disease (CHD) outcome provided by external validation studies included in the meta-analyses.**
(DOCX)

**S2 Table. List of excluded articles and categories for the exclusion.**
(DOCX)

**S3 Table. Journals that published the included studies of the Framingham Wilson coronary heart disease risk rule.**
(DOCX)

**S4 Table. Studies that derived a new cardiovascular clinical prediction rule using the poor performance of the Framingham Wilson coronary heart disease risk rule in the external validation study as one of the justifications.**
(DOCX)

**S1 Appendix. Methods for meta-analysis of predicted and observed (P/O) ratio and $c$ statistic.**
(DOCX)

**S2 Appendix. PRISMA checklist.**
(DOCX)

**S3 Appendix. List of the included articles of the Framingham Wilson coronary heart disease risk rule.**
(DOCX)

**S1 Dataset.**
(XLSX)

## Author Contributions

**Conceptualization:** Jong-Wook Ban, Lucy Abel, Richard Stevens, Rafael Perera.

**Data curation:** Jong-Wook Ban.

**Formal analysis:** Jong-Wook Ban, Lucy Abel.

**Investigation:** Jong-Wook Ban, Lucy Abel, Richard Stevens, Rafael Perera.

**Methodology:** Jong-Wook Ban.

**Project administration:** Jong-Wook Ban.

**Resources:** Jong-Wook Ban.

**Supervision:** Richard Stevens, Rafael Perera.

**Writing – original draft:** Jong-Wook Ban.

**Writing – review & editing:** Jong-Wook Ban, Lucy Abel, Richard Stevens, Rafael Perera.

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
