## [Decision Letter · Decision Letter 0]

5 Nov 2022

PONE-D-22-18030Research inefficiencies in external validation studies of the Framingham Wilson coronary heart disease risk rule: A systematic reviewPLOS ONE

Dear Dr. Ban,

Thank you for submitting your manuscript to PLOS ONE. After careful consideration, we feel that it has merit but does not fully meet PLOS ONE’s publication criteria as it currently stands. Therefore, we invite you to submit a revised version of the manuscript that addresses the points raised during the review process.

ACADEMIC EDITOR:We are interested in reviewing a revised manuscript after addressing the comments and concerns raised by the reviewers below.

We look forward to receiving your revised manuscript.

Kind regards,

Fares Alahdab

Academic Editor

PLOS ONE

Journal Requirements:

2. Please ensure that your database search is up to date. We note that it was last completed in 2020 and per our internal policy we cannot consider systematic reviews which were not updated for more that 12 months. Thank you for your attention to this request

Reviewers' comments:

Reviewer's Responses to Questions

**Comments to the Author**

1. Is the manuscript technically sound, and do the data support the conclusions?

Reviewer #1: Partly

Reviewer #2: Yes

2. Has the statistical analysis been performed appropriately and rigorously? 

Reviewer #1: Yes

Reviewer #2: No

3. Have the authors made all data underlying the findings in their manuscript fully available?

Reviewer #1: Yes

Reviewer #2: Yes

4. Is the manuscript presented in an intelligible fashion and written in standard English?

Reviewer #1: Yes

Reviewer #2: Yes

5. Review Comments to the Author

Reviewer #1: Research inefficiencies in external validation studies of the Framingham Wilson coronary heart disease risk rule: A systematic review

The external validation of clinical prediction rules is a key issue in their importance moving forward. But the relative inconsistency of how to do such studies and what to do with them is holding the field back. This study was a systematic review to assess how well that is going. There’s a lot of good work in here including a systematic review with good data extraction.

The writing is generally good though they should follow standard structure at touch more. For example, the methods section is under-structured and I was unclear what the outcome measures were

Unfortunately, the two main criteria the study was looking at just didn’t strike me as fair or valuable. They assessed if (1) the new model updated the original and (2) if they assessed overall performance.

I’m unconvinced by their belief that any new model should make a new parameterization of the old. What do we do with a new parameterization of an old model? We have so many and applying them is unclear. We have dozens of new models based on old ones, the new ones aren’t externally valid either (and won’t be) because the differences between populations are real, not a sign of flaws that the next paper will correct. So I’m unclear what we’d do with the information and why.

Their finding that model assessment is unclear and inconsistent is more valuable, though also problematic. Finding that CPRs don’t present C-statistics is bad and interesting. It fits my experience as well. But the calibration assessment, which the authors focus on, is much less clear. The authors emphasize the predicted to observed ratio. While I like this ratio also, it’s an an atypical measure that isn’t even mentioned in the TRIPOD elaboration paper. Expecting it to be presented isn’t fair to current articles. To me the larger problem here is that it's legitimately unclear what calibration measures should be presented, not that papers are doing a poor job presenting those results.

I suppose if it were me I would focus on measures of model assessment, de-emphasize the P:O ratio and simply show the # of papers that presented no measures of discrimination and no measure of calibration. I think everyone would agree that without those, an article is missing key features.

Reviewer #2: The manuscript is well written and well structured. It focuses on an important area of research relevant to the quality of clinical prediction models. More specifically, it focuses on research inefficiencies in external validation studies of the Framingham Wilson coronary heart disease risk rule. Findings from this paper could be generalized to external validation studies in other domains.

The manuscript could benefit from:

- better consistency of study findings to align with objectives

- elaborating why only two areas of research inefficiencies were focused on versus including other TRIPOD items

- updating the search to include more recent publications

- realigning/restructuring presentation of results

More detailed comments are stated below.

ABSTRACT

- The authors indicate that 39 external validation studies were identified. Later in the manuscript the authors present that 97 studies were included. These additional studies should also be presented in the abstract to avoid confusion (they are also likely external validation studies)

INTRODUCTION

- The introduction is clear but would be helpful to readers to know when the Framingham Wilson CHD paper was published. This will help the reader to make a connection with the TRIPOD adherence depending on which year they were published - especially useful for the discussion section.

METHODS

- In the methods you mentioned "We evaluated a CPR for coronary heart disease (CHD) risk derived by Wilson et al. (24) using the Framingham Heart Study cohort in 1998." Perhaps you meant to be evaluating external validation studies of that CPR?

- Search was updated in June 2020. That was more than two years ago so there are likely other (and possibly improved) external validations of the Framingham Wilson CPR? I suggest updating it.

- Scopus was used. Embase is known to include additional citations. Might be helpful to check with an information librarian if it is worth including other databases.

- In methods you indicate the search was not limited to any language. Did you find any in other languages? Did you end up needing a translator for them?

- Under Data Analysis, you indicate "(c) performed inadequately but did not update the CPR." I assume these are the studies that are considered inefficient. Could it be that they just wanted to externally validate the study and not update or derive a new rule? Or are these studies that ended up deriving a new rule? Please refer to TRIPOD and other studies which categorize external validation studies as with and without updating.

- Authors could look into using the Critical Appraisal and Data Extraction for Systematic Reviews of Prediction Modelling Studies (CHARMS) to extract and present more comprehensive issues with external validation studies. For example, authors have referenced the TRIPOD guideline. There are several other items that could be included in this study including TRIPOD items 4-12.

RESULTS

- In the Introduction you state that "For example, a systematic review found that the Framingham Wilson coronary heart disease (CHD) risk rule was the most frequently validated cardiovascular CPR with 89 external validation studies (22)." Please see above but you report that you have identified only 39 external validation studies.

- In connection with above point, I think there are more external validation studies than 39. It might be due to how you define external validation studies versus the other 58 studies. The other studies are also likely external validation studies but with reporting pitfalls (i.e. they are possibly not reporting that they are externally validating). Reporting pitfalls are common with derivation and validation studies and hence TRIPOD was developed as a guidance.

- I understand that 26 out of 29 studies concluded the Framingham Wilson CHD risk rule performed poorly but did not update it. Some studies have no plan to either create a new rule nor update it but rather just want to see how well the rule performs on their patients, as is - these should be their own group. However, among the studies that either derived a new rule or updated the existing one, what proportions were deriving a new rule (vs updating it)?

- In addition to above point, for the studies that updated the rule, how many considered less extensive methods prior to trying more extensive methods to update the prediction rule?

- The results in the Tables would need to be reorganized once the studies are combined. Perhaps to list all as external validation studies but show a column that indicates if the authors reported their objectives clearly. Also, put more emphasis on results answering the objectives.

DISCUSSION

- Findings from this study on deficiencies of external validation studies is a common issue in other domains. Would be useful to readers to compare with other systematic reviews in other domains.

- The authors indicate "Even when authors of the external validation studies reported these performance measures of the Framingham Wilson CHD risk rule, most did not adhere to the recommendations from the TRIPOD statement." As with the Introduction it would be good to know how many of the external validation studies were published prior to the TRIPOD statement -- a limitations to adhering to TRIPOD statement.

6. PLOS authors have the option to publish the peer review history of their article (what does this mean?). If published, this will include your full peer review and any attached files.

Reviewer #1: No

Reviewer #2: **Yes: **Kasim Abdulaziz

---

## [Author Response · Author response to Decision Letter 0]

8 Mar 2023

Centre for Evidence-Based Medicine and Department for Continuing Education

University of Oxford

Oxford, United Kingdom

10 February 2023

Fares Alahdab

Academic Editor

PLOS ONE

Dear Dr Alahdab, 

Thank you very much for inviting us to revise our manuscript. We are pleased to submit the revised version of our manuscript (PONE-D-22-18030) titled “Research inefficiencies in external validation studies of the Framingham Wilson coronary heart disease risk rule: A systematic review.”

We would also like to thank the peer reviewers for evaluating our manuscript and providing constructive feedback. We confirm that our electronic database searches have been updated as of December 2022 and our revised manuscript meets the journal’s style requirements. We tried our best to address the reviewers’ comments by explaining our rationale better, describing methods clearer, and discussing the results considering contexts.

For each question raised by the reviewers, which is highlighted in bold italic characters, we provided our response followed by any revision(s) made to the manuscript in bullet point (line in marked up copy). We look forward to hearing back from you soon. 

With best wishes, 

Jong-Wook Ban, MD, MSc, DPhil 

Reviewer #1

The external validation of clinical prediction rules is a key issue in their importance moving forward. But the relative inconsistency of how to do such studies and what to do with them is holding the field back. This study was a systematic review to assess how well that is going. There’s a lot of good work in here including a systematic review with good data extraction.

1. The writing is generally good though they should follow standard structure at touch more. For example, the methods section is under-structured and I was unclear what the outcome measures were.

We appreciate the reviewer’s comment. We restructured the methods section by adding a section titled “Outcome measures” and revised the contents of the data analysis section to help readers understand methods better. 

• Line 230 - 236: “Research inefficiency occurs when research studies fail to “contribute to knowledge or to practice and policy,” to their maximum potential (22, 37). In our study, we assessed two types of inefficiency among external validation studies of the Framingham Wilson CHD risk rule. Firstly, we evaluated the inefficiency from external validation studies that do not attempt to update the Framingham Wilson CHD risk rule when it performs poorly; thus, missing the opportunity to create a CHD risk rule that could be useful for populations similar to the one used in the external validation.”

• Line 238 – 343: “Secondly, we assessed the inefficiency from external validation studies that are potentially eligible to be included in a meta-analysis and contribute to understanding the performance of the Framingham Wilson CHD risk rule, but fail to do so because either a P/O ratio or c statistic is not reported or cannot be obtained from provided data. We chose to evaluate the P/O ratio because existing systematic reviews (38-40) of the Framingham Wilson CHD risk rule invariably meta-analyzed the P/O ratio to summarize calibration.”

• Line 269 – 274: “We presented the proportion of external validation studies in which authors concluded that the Framingham Wilson CHD risk rule (a) performed adequately, (b) performed inadequately and updated the CPR, and (c) performed inadequately but did not update the CPR. We also calculated the proportion of eligible external validation studies that could not be included in the meta-analysis of P/O ratio or c statistic because a relevant performance measure was not reported or could not be estimated from provided data.”

• Line 276 – 282: “Additionally, we conducted post hoc citation analyses to evaluate the potential impact of external validation studies in which authors concluded that the Framingham Wilson CHD risk rule performed inadequately but did not update it. In Scopus, we ran forward citation searches of these external validation studies, which were updated in December 2022. We searched for studies that derived a new cardiovascular CPR while using the Framingham Wilson CHD risk rule's poor performance as one of the justifications and citing one of these external validation studies.”

2. Unfortunately, the two main criteria the study was looking at just didn’t strike me as fair or valuable. They assessed if (1) the new model updated the original and (2) if they assessed overall performance.

I’m unconvinced by their belief that any new model should make a new parameterization of the old. What do we do with a new parameterization of an old model? We have so many and applying them is unclear. We have dozens of new models based on old ones, the new ones aren’t externally valid either (and won’t be) because the differences between populations are real, not a sign of flaws that the next paper will correct. So I’m unclear what we’d do with the information and why.

We revised the following paragraphs in the introduction section to more clearly explain why updating the sub-optimally performed CPR might be more efficient use of external validation studies. 

• Line 64 - 75: “However, CPRs often perform unsatisfactorily when applied to populations different from the ones they are derived, in external validation studies (8-10), which might be due to inflated accuracy from suboptimal design and methods used to derive the CPRs or differences in populations and settings between derivation and external validation studies (4, 11).”

• Line 77 - 85: “Sometimes, authors of external validation studies declare that the CPRs, when they perform poorly, are not generalizable (12-14). There are several disadvantages for this approach. When no valid CPR exists for the population, clinicians cannot make decisions guided by a CPR until a new accurate CPR is created by often repeating the entire derivation process from ground up, because using poorly performing CPR can lead to decisions harmful to patients (15). Moreover, this means that the knowledge accumulated while deriving the CPR is simply wasted (11, 16). Further, this approach might promote the proliferation of many redundant CPRs for the same problem (11, 16) because the poor performance of CPR is one of the most commonly stated justifications by authors who derive a new cardiovascular CPR (17).”

• Line 87 - 97: “Alternatively, authors could attempt to update the CPR while conducting the external validation study when it performs poorly (3, 11, 16, 18). The performance of the CPR could improve simply by adjusting the calibration intercept or by re-estimating coefficients of predictor variables (11, 19-21). To the contrary to dismissing the CPR that performs poorly in external validation, previous research and evidence accrued is efficiently used when the CPR is updated (11, 16, 22). Although the updated CPR still should be externally validated, with this approach, it might be unnecessary for researchers to create a new CPR just for this population by rerunning the entire derivation process (11, 16). For example, the REGICO function was created by calibrating the Framingham Wilcon CHD risk rule in Spanish cohort (23), which has been successfully validated in various Spanish cohorts (24-26), and adopted by Spanish clinical practice guidelines (27, 28).

3. Their finding that model assessment is unclear and inconsistent is more valuable, though also problematic. Finding that CPRs don’t present C-statistics is bad and interesting. It fits my experience as well. But the calibration assessment, which the authors focus on, is much less clear. The authors emphasize the predicted to observed ratio. While I like this ratio also, it’s an atypical measure that isn’t even mentioned in the TRIPOD elaboration paper. Expecting it to be presented isn’t fair to current articles. To me the larger problem here is that it's legitimately unclear what calibration measures should be presented, not that papers are doing a poor job presenting those results.

Calibration of a clinical prediction rule can be presented in several ways as shown in Table 2 of our manuscript. To our knowledge, existing systematic reviews of the Framingham Wilson CHD risk rule that meta-analyzed a calibration measure almost always focused on the P/O ratio. Therefore, we evaluated whether external validation studies of the Framingham Wilson CHD risk rule either reported a P/O ratio or provided sufficient data to calculate it; and could contribute to understanding the performance of the Framingham Wilson CHD risk rule, to their maximum potential. We revised the methods section to make this rationale clear to readers. 

• Line 238 - 243: “Secondly, we assessed the inefficiency from external validation studies that are potentially eligible to be included in a meta-analysis and contribute to understanding the performance of the Framingham Wilson CHD risk rule, but fail to do so because either a P/O ratio or c statistic is not reported or cannot be obtained from provided data. We chose to evaluate the P/O ratio because existing systematic reviews (38-40) of the Framingham Wilson CHD risk rule invariably meta-analyzed the P/O ratio to summarize calibration.”

4. I suppose if it were me I would focus on measures of model assessment, de-emphasize the P:O ratio and simply show the # of papers that presented no measures of discrimination and no measure of calibration. I think everyone would agree that without those, an article is missing key features.

We revised the results section to highlight the number and proportion of external validation studies that failed to report any discrimination and calibration measures. 

• Line 316 - 486: “Performance measures of the Framingham Wilson CHD risk rule reported by the included studies are summarised in Table 2. Of 39 studies that aimed to externally validate the Framingham Wilson CHD risk rule, 17 (43.6%) did not reported any calibration measure. Only 14 (35.9%) studies reported calibration according to the recommendations from the TRIPOD statement: 5 (12.8%) studies with both calibration table and plot, 8 (20.5%) studies with a calibration plot only, one (2.6%) study with a calibration table only. A measure of discrimination was not reported in a minority (11 of 39, 28.2%) of studies that aimed to externally validate the Framingham Wilson CHD risk rule. However, c statistic was reported with a 95% confidence interval only in 16 (41.0%) external validation studies, adhering to the recommendation in the TRIPOD statement.”

Reviewer #2: 

The manuscript is well written and well structured. It focuses on an important area of research relevant to the quality of clinical prediction models. More specifically, it focuses on research inefficiencies in external validation studies of the Framingham Wilson coronary heart disease risk rule. Findings from this paper could be generalized to external validation studies in other domains.

The manuscript could benefit from:

- better consistency of study findings to align with objectives

- elaborating why only two areas of research inefficiencies were focused on versus including other TRIPOD items

- updating the search to include more recent publications

- realigning/restructuring presentation of results

More detailed comments are stated below.

ABSTRACT

1. The authors indicate that 39 external validation studies were identified. Later in the manuscript the authors present that 97 studies were included. These additional studies should also be presented in the abstract to avoid confusion (they are also likely external validation studies).

We appreciate the reviewer’s comment. We revised the abstract to avoid confusion by adding a sentence to present 98 included studies that evaluated the Framingham Wilson coronary heart disease (CHD) risk rule. Please note that we wish to keep the distinction between two types of include studies. We explained our rationale for this in our response to the reviewer’s comment 10.

• Line 34 - 35: “We identified 98 studies that evaluated the Framingham Wilson CHD risk rule; 40 of which were external validation studies.”

INTRODUCTION

2. The introduction is clear but would be helpful to readers to know when the Framingham Wilson CHD paper was published. This will help the reader to make a connection with the TRIPOD adherence depending on which year they were published - especially useful for the discussion section.

We revised the introduction section and explained when the Framingham Wilson CHD risk rule was published. 

• Line 147 - 149: “… a systematic review found that the Framingham Wilson coronary heart disease (CHD) risk rule, which was published in 1998, was the most frequently validated cardiovascular CPR with 89 external validation studies …”

METHODS

3. In the methods you mentioned "We evaluated a CPR for coronary heart disease (CHD) risk derived by Wilson et al. (24) using the Framingham Heart Study cohort in 1998." Perhaps you meant to be evaluating external validation studies of that CPR?

We changed the paragraph to make it clear that we evaluated external validation studies of the Framingham Wilson CHD risk rule. 

• Line 160 - 162: “We evaluated external validation studies of a CPR for coronary heart disease (CHD) risk derived by Wilson et al. (24) using the Framingham Heart Study cohort. Published in 1998, the Framingham Wilson CHD risk rule estimates the 10-year risk of CHD …”

4. Search was updated in June 2020. That was more than two years ago so there are likely other (and possibly improved) external validations of the Framingham Wilson CPR? I suggest updating it.

We updated the forward citation search as of December 2022. In addition, the searches for the additional analysis were also updated. We revised the abstract, methods, and results section. Also, figure and tables have been updated accordingly. 

• Line 35 - 47: “Of these 40 studies, 27 (67.5%) concluded the Framingham Wilson CHD risk rule performed poorly but did not update it. Of 23 external validation studies conducted with data that could be included in meta-analyses, 13 (56.5%) could not fully contribute to the meta-analyses ...”

• Line 173 - 174: “This initial search was updated in June 2020 and December 2022.”

• Line 278 - 279: “In Scopus, we ran forward citation searches of these external validation studies, which were updated in December 2022.”

• Line 293 - 294: “A total of 98 studies that evaluated the Framingham Wilson CHD risk rule from 84 publications were included, Fig 1.”

• Line 300 - 302: “For 40 (40.8%) of 98 included studies, one of the aims was externally validating the Framingham Wilson CHD risk rule. For 58 (59.2%) studies …”

• Line 673 – 675: “Of 40 external validation studies of the Framingham Wilson CHD risk rule, authors of 27 external validation studies (67.5%) concluded that the performance was inadequate but did not update the CPR.”

• Line 701 – 704: “In a post hoc citation analysis, a total of 1,341 references were found in forward citation searches of 26 external validation studies that concluded the performance of the Framingham Wilson CHD risk rule was inadequate but did not update it. From these references, we found 20 studies …”

• Line 704: “A total of 44 studies were included in the meta-analysis of the P/O ratio: 15 studies …”

• Line 738 - 740: “A total of 24 studies were included in the meta-analysis of c statistic: 16 studies with an explicit aim of externally validating the Framingham Wilson CHD risk rule and 8 studies where a c statistic could be obtained from data.”

• Line 765 – 767: “Of 40 studies that aimed to externally validate the Framingham Wilson CHD risk rule, 23 studies were conducted in the USA and Europe: therefore could have been included in the meta-analyses as presented in Table 3.”

• Line 808 – 810: “In summary, of 23 studies that aimed to externally validate the Framingham Wilson CHD risk rule, only ten (43.5%) contributed to understanding both the summary P/O ratio and c statistic in eligible meta-analyses. On the other hand, 13 (56.5%) studies that …”

5. Scopus was used. Embase is known to include additional citations. Might be helpful to check with an information l

---

## [Decision Letter · Decision Letter 1]

16 May 2023

PONE-D-22-18030R1Research inefficiencies in external validation studies of the Framingham Wilson coronary heart disease risk rule: A systematic reviewPLOS ONE

Dear Dr. Ban,

Thank you for submitting your manuscript to PLOS ONE. After careful consideration, we feel that it has merit but does not fully meet PLOS ONE’s publication criteria as it currently stands. Therefore, we invite you to submit a revised version of the manuscript that addresses the points raised during the review process.

We look forward to receiving your revised manuscript.

Kind regards,

Miquel Vall-llosera Camps

Senior Editor

PLOS ONE

Additional Editor Comments:

It was considered necessary to invite additional reviewers to provide assessment. The reviewers have raised additional concerns about your study that need to be addressed.

Reviewers' comments:

Reviewer's Responses to Questions

**Comments to the Author**

1. If the authors have adequately addressed your comments raised in a previous round of review and you feel that this manuscript is now acceptable for publication, you may indicate that here to bypass the “Comments to the Author” section, enter your conflict of interest statement in the “Confidential to Editor” section, and submit your "Accept" recommendation.

Reviewer #1: All comments have been addressed

Reviewer #3: All comments have been addressed

Reviewer #4: (No Response)

Reviewer #5: All comments have been addressed

2. Is the manuscript technically sound, and do the data support the conclusions?

Reviewer #1: Yes

Reviewer #3: Yes

Reviewer #4: Yes

Reviewer #5: Yes

3. Has the statistical analysis been performed appropriately and rigorously? 

Reviewer #1: Yes

Reviewer #3: Yes

Reviewer #4: I Don't Know

Reviewer #5: Yes

4. Have the authors made all data underlying the findings in their manuscript fully available?

Reviewer #1: Yes

Reviewer #3: Yes

Reviewer #4: Yes

Reviewer #5: No

5. Is the manuscript presented in an intelligible fashion and written in standard English?

Reviewer #1: Yes

Reviewer #3: Yes

Reviewer #4: Yes

Reviewer #5: Yes

6. Review Comments to the Author

Reviewer #1: I appreciate the response to my comments. I do still find the methods section confusing, primarily due to a tendency to not follow their outline.

1. The “outcomes” section does not state the primary outcome. Research inefficiency is a nice theoretical construct, but it’s not an outcome measure. I believe it’s currently in data analysis (I think it’s P/O and c-stat), but I’m still not positive and it’s in the wrong place. “Our primary outcome is the reporting of P/O ratio and c-statistic in the valiation study.” Or something.

2. The first sentence of data analysis is actually about inclusion criteria, right? Shouldn’t that be in study selection checklist?

3. The first paragraph of Information Sources is a description of the Wilson score, which is not an information source and should probably be in the introduction.

Reviewer #3: The R1 revision of the manuscript and the responses have addressed the previous reviewers' comments. While I agree with a previous reviewer that the meta-analyses of P/O ratio as an assessment of calibration is less clear compared with meta-analyses of C-statistics as an assessment of discrimination, there is probably little researchers can improve based on aggregated data in the framework of systematic review + meta-analyses (unless individual level data could be used).

I have two suggestions stated below.

Lines 283-308 and texts that corresponds to Fig 3-A, 3-B and Fig 4:

Although under "Study Selection" the authors stated that the cardiovascular outcome evaluated was "either CHD (a composite of angina pectoris, acute myocardial infarction, coronary insufficiency, and CHD death) or hard CHD (a composite of acute myocardial infarction, coronary insufficiency, and CHD death)", the authors did not mention what exact definition of the outcome has been used for each study (for example, in the form of stratified or sub analyses) in the meta-analyses of P/O ratio (Fig 3-A, 3-B) and C-statistics (Fig 4). The definition of reported cardiovascular outcome differs from study to study, for example, "death from CHD" is quite different from "total or hard CHD". I suggest that another supplemental material could list the definition of cardiovascular outcome(s) by study that correspond to extracted data in this review and the meta-analyses of P/O ratio and C-statistics.

Figures 3-A and 3-B: It would be better if two columns of numbers, "predicted n/N" and "observed n/N" that have been extracted from the studies, can be displayed on the forest plots, same as references 38 (Brindle et al) and 39 (Eichler et al). The numerators must correspond to the specific definition of reported cardiovascular outcome in the supplemental material (mentioned above).

Reviewer #4: I feel this is a very well written article. It illustrates some important shortcomings in current external validation studies. The use of the TRIPOD reporting guidelines and the value of reducing research inefficiencies and waste are important messages that are well illustrated by this article. I agree that by not following TRIPOD reporting guidelines, there are lost opportunities to contribute to further understanding of CPRs. As noted, when a CPR performs well, clinicians more likely to have confidence in it and use it – there is a need to avoid just dismissing it if doesn’t perform well. The authors have made an excellent choice in selecting the Framingham Wilson coronary heart disease (CHD) risk rule, given its longstanding, wide use in clinical practice.

I thought the method for identifying studies to be logical and appropriate. I would suggest the authors justify their use of Scopus over other forward-citation tools, as the results differ by source. An initial check of three sources showed that Scopus does produce the largest number of results (7442), followed by Web of Science (6717) and PubMed (2666). Although there will be significant overlap across these sources, I believe there will still be unique results from each, so it is possible some potentially relevant records were missed due to a full reliance on Scopus. I also found the inclusion criteria logical and would commend the authors for considering the availability of performance measures in the absence of a completed external validation. I thought the post hoc analyses of group C in which it was noted the Framingham RR performed inadequately but the authors did not update CPR, particularly interesting.

I would like to know why only studies in USA and Europe were considered and believe this needs more explanation in the text. I would think Australia, New Zealand and Canada would also have a comparable CVD burden.

I am unable to comment on the statistical methods or results. However, the overall conclusions were sound and quite compelling. Why dismiss existing tools and replace them rather than improve and update an existing tool and contribute to the associated evidence base? I like the reference to the power of journals in encouraging authors to justify their work and to undertake updates of a CPR when it performs poorly.

This is a very practical and useful article.

Reviewer #5: Overall, the paper addresses an important topic by investigating the research inefficiencies in studies evaluating the Framingham Wilson coronary heart disease (CHD) risk rule. The authors systematically review the literature, including 98 studies, and focus on two types of research inefficiencies: failure to update the CHD risk rule and inability to contribute to understanding the overall performance of the rule.

The paper provides valuable insights into the extent of inefficiencies in the research process, highlighting that many studies did not update the risk rule despite finding its performance inadequate. Additionally, a significant number of studies did not contribute to eligible meta-analyses due to the lack of reporting relevant performance measures or providing data for their estimation. The paper is well-structured and presents the results in a clear and organized manner, using tables and figures to help readers understand the findings. The meta-analyses conducted by the authors also add to the strength of the study, as they synthesize the data from multiple sources to provide a more comprehensive understanding of the risk rule's performance.

In particular, the methodology used in this SR showed certain strengths:

• Systematic review: The authors conducted a systematic review of the literature, following the PRISMA guidelines. This approach helps ensure a comprehensive and unbiased selection of studies for inclusion, enhancing the validity of the paper's conclusions.

• Inclusion of many studies: The paper included a total of 98 studies, providing a broad and diverse sample to assess the research inefficiencies related to the Framingham Wilson CHD risk rule.

• Use of meta-analyses: The authors conducted meta-analyses to synthesize data from multiple studies, providing more precise estimates of the risk rule's performance across different populations and settings. This approach adds to the strength of the study, as it enables a more comprehensive understanding of the risk rule's overall performance.

• Assessment of study characteristics and reporting: The paper thoroughly evaluates the characteristics of the included studies, as well as their reporting of performance measures. This assessment helps identify areas where research practices could be improved to address the observed inefficiencies.

However, there was Lack of exploration of reasons behind inefficiencies: The paper does not delve deeply into the underlying reasons for the observed research inefficiencies. Understanding these factors could help inform more specific recommendations for improving research practices in this area. The authors could conduct further analyses or interviews with researchers to better understand the factors contributing to the observed research inefficiencies. This could help inform targeted recommendations for improving research practices.

The writing quality of the paper is generally good, with clear language and appropriate terminology. The organization of the paper follows a logical structure, with a clear introduction, methods, results, and discussion sections. The clinical usefulness of the paper is evident, as it informs researchers and clinicians about the potential limitations of the risk rule and the need for updating it.

In conclusion, the paper provides a valuable contribution to the literature by highlighting the research inefficiencies in studies evaluating the Framingham Wilson CHD risk rule. This could potentially lead to improvements in future research and help enhance the overall quality and utility of clinical prediction rules in the field of cardiovascular disease.

7. PLOS authors have the option to publish the peer review history of their article (what does this mean?). If published, this will include your full peer review and any attached files.

Reviewer #1: No

Reviewer #3: No

Reviewer #4: No

Reviewer #5: No

---

## [Author Response · Author response to Decision Letter 1]

28 Jun 2023

Reviewer #1: I appreciate the response to my comments. I do still find the methods section confusing, primarily due to a tendency to not follow their outline.

1. The “outcomes” section does not state the primary outcome. Research inefficiency is a nice theoretical construct, but it’s not an outcome measure. I believe it’s currently in data analysis (I think it’s P/O and c-stat), but I’m still not positive and it’s in the wrong place. “Our primary outcome is the reporting of P/O ratio and c-statistic in the valiation study.” Or something.

We added the following sentences to the outcome measures sections to define our main outcome measures clearly.

• Line 257-259: “Therefore, one of our principal outcome measures was the proportion of external validation studies that concluded the Framingham Wilson CHD risk rule performed inadequately but did not update the CPR.”

• Line 264-267: “Thus, our other main outcome measure was the proportion of eligible external validation studies that could not be included in the meta-analysis of P/O ratio or c statistic due to the insufficient reporting of relevant performance measures.”

2. The first sentence of data analysis is actually about inclusion criteria, right? Shouldn’t that be in study selection checklist?

We apologize for the confusion. The first sentence is not about inclusion criteria, but about which studies are used in the meta-analysis. To make the method clearer to readers, we revised the section. Firstly, we explained how we analyzed the inefficiency from external validation studies that do not update the Framingham Wilson CHD risk rule when it performs poorly. Then, we described how we measured the inefficiency from external validation studies that are potentially eligible to be included in a meta-analysis but fail to do so due to the insufficient reporting of relevant performance measures.

• Line 272-275: “Using all included external validation studies, we calculated the proportion of studies in which authors concluded that the Framingham Wilson CHD risk rule (a) performed adequately, (b) performed inadequately and updated the CPR, and (c) performed inadequately but did not update the CPR.” 

• Line 277-280: For studies conducted in the USA and European geographic regions with comparable cardiovascular disease burden, the prevalence of risk factors, and health care environment (41): the UK, Northern Europe, Western Europe, and Southern Europe, we meta-analysed the Predicted/Observed (P/O) event ratio and c statistic of the Framingham Wilson CHD risk rule. 

• Line 286-288: We presented the proportion of eligible external validation studies that could not be included in the meta-analysis of P/O ratio or c statistic because a relevant performance measure was not reported or could not be estimated from provided data.

3. The first paragraph of Information Sources is a description of the Wilson score, which is not an information source and should probably be in the introduction.

We moved the paragraph to the introduction section and revised it to fit the flow. 

• Line 149-158: “In 1998, Wilson et al. (33) derived a CPR for coronary heart disease (CHD) risk using the Framingham Heart Study cohort. The Framingham Wilson CHD risk rule estimates the 10-year risk of CHD (composite of angina pectoris, acute myocardial infarction, coronary insufficiency, and CHD death) and hard CHD (all CHD outcomes except for angina pectoris) with the following predictors: age, cigarette use, diabetes mellitus status, total and high-density lipoprotein cholesterol categories, and blood pressure categories (33). Systematic reviews (31, 34) have shown that more external validation studies for the Framingham Wilson CHD risk rule have been conducted than any other cardiovascular CPR. For example, a systematic review found that the Framingham Wilson CHD risk rule was the most frequently validated cardiovascular CPR with 89 external validation studies (31).”

Reviewer #3: The R1 revision of the manuscript and the responses have addressed the previous reviewers' comments. While I agree with a previous reviewer that the meta-analyses of P/O ratio as an assessment of calibration is less clear compared with meta-analyses of C-statistics as an assessment of discrimination, there is probably little researchers can improve based on aggregated data in the framework of systematic review + meta-analyses (unless individual level data could be used).

I have two suggestions stated below.

4. Lines 283-308 and texts that corresponds to Fig 3-A, 3-B and Fig 4: Although under "Study Selection" the authors stated that the cardiovascular outcome evaluated was "either CHD (a composite of angina pectoris, acute myocardial infarction, coronary insufficiency, and CHD death) or hard CHD (a composite of acute myocardial infarction, coronary insufficiency, and CHD death)", the authors did not mention what exact definition of the outcome has been used for each study (for example, in the form of stratified or sub analyses) in the meta-analyses of P/O ratio (Fig 3-A, 3-B) and C-statistics (Fig 4). The definition of reported cardiovascular outcome differs from study to study, for example, "death from CHD" is quite different from "total or hard CHD". I suggest that another supplemental material could list the definition of cardiovascular outcome(s) by study that correspond to extracted data in this review and the meta-analyses of P/O ratio and C-statistics.

As suggested by the reviewer, we provided the outcome definitions from studies that were included in the meta-analyses in a supplemental table. 

• Line 281-282: “The definitions of CHD from external validation studies included in the meta-analyses are summarized in S1 Table.”

5. Figures 3-A and 3-B: It would be better if two columns of numbers, "predicted n/N" and "observed n/N" that have been extracted from the studies, can be displayed on the forest plots, same as references 38 (Brindle et al) and 39 (Eichler et al). The numerators must correspond to the specific definition of reported cardiovascular outcome in the supplemental material (mentioned above).

We revised Figure 3-A and 3-B to add the numbers of predicted and observed events. 

Reviewer #4: I feel this is a very well written article. It illustrates some important shortcomings in current external validation studies. The use of the TRIPOD reporting guidelines and the value of reducing research inefficiencies and waste are important messages that are well illustrated by this article. I agree that by not following TRIPOD reporting guidelines, there are lost opportunities to contribute to further understanding of CPRs. As noted, when a CPR performs well, clinicians more likely to have confidence in it and use it – there is a need to avoid just dismissing it if doesn’t perform well. The authors have made an excellent choice in selecting the Framingham Wilson coronary heart disease (CHD) risk rule, given its longstanding, wide use in clinical practice.

6. I thought the method for identifying studies to be logical and appropriate. I would suggest the authors justify their use of Scopus over other forward-citation tools, as the results differ by source. An initial check of three sources showed that Scopus does produce the largest number of results (7442), followed by Web of Science (6717) and PubMed (2666). Although there will be significant overlap across these sources, I believe there will still be unique results from each, so it is possible some potentially relevant records were missed due to a full reliance on Scopus. 

As advised, we further explained the reasons for choosing Scopus in the information source and search for external validation studies section. Also, we acknowledge the possibility that relevant records were missed because we used a single data source in the strength and limitation section. 

• Line 174-178: “We conducted our forward citation search using Scopus as a citation index because we have previously demonstrated that this method could efficiently and systematically identify CPR studies (35, 36). Further, for citation searches in the health science field, Scopus tends to produce more robust results than Web of Science while minimally missing unique references (37, 38).”

• Line 968-972: “On the other hand, we only relied on Scopus to conduct our forward citation search. Although the use of a single citation index is a very common practice (81), and Scopus performs well compared with Web of Science in the health science field (37, 38), searching multiple sources might have been more ideal to ensure none of the potentially relevant references was missed.”

I also found the inclusion criteria logical and would commend the authors for considering the availability of performance measures in the absence of a completed external validation. I thought the post hoc analyses of group C in which it was noted the Framingham RR performed inadequately but the authors did not update CPR, particularly interesting.

7. I would like to know why only studies in USA and Europe were considered and believe this needs more explanation in the text. I would think Australia, New Zealand and Canada would also have a comparable CVD burden.

Interpreting random-effects meta-analyses that combine the results of external validation studies from multiple countries, over different time periods, with different eligibility criteria can be difficult. Ideally, a meta-analysis of external validation studies should include studies from “plausibly related populations.” (1) However, it is not always easy to clearly determine the plausibility. Rather, we often need to evaluate how plausible it is that populations are related, which requires an epidemiological understanding about the topic but also subjective judgement. In our study, we took a conservative approach and only meta-analyzed the studies from Europe and grouped them into studies form the UK, Northern Europe, Western Europe, and Southern Europe. Nevertheless, there were only one study from Australia and one study from Canada; including these two studies would not have changed the overall conclusion of our study. 

I am unable to comment on the statistical methods or results. However, the overall conclusions were sound and quite compelling. Why dismiss existing tools and replace them rather than improve and update an existing tool and contribute to the associated evidence base? I like the reference to the power of journals in encouraging authors to justify their work and to undertake updates of a CPR when it performs poorly.

This is a very practical and useful article.

Reviewer #5: Overall, the paper addresses an important topic by investigating the research inefficiencies in studies evaluating the Framingham Wilson coronary heart disease (CHD) risk rule. The authors systematically review the literature, including 98 studies, and focus on two types of research inefficiencies: failure to update the CHD risk rule and inability to contribute to understanding the overall performance of the rule.

The paper provides valuable insights into the extent of inefficiencies in the research process, highlighting that many studies did not update the risk rule despite finding its performance inadequate. Additionally, a significant number of studies did not contribute to eligible meta-analyses due to the lack of reporting relevant performance measures or providing data for their estimation. The paper is well-structured and presents the results in a clear and organized manner, using tables and figures to help readers understand the findings. The meta-analyses conducted by the authors also add to the strength of the study, as they synthesize the data from multiple sources to provide a more comprehensive understanding of the risk rule's performance. In particular, the methodology used in this SR showed certain strengths:

• Systematic review: The authors conducted a systematic review of the literature, following the PRISMA guidelines. This approach helps ensure a comprehensive and unbiased selection of studies for inclusion, enhancing the validity of the paper's conclusions.

• Inclusion of many studies: The paper included a total of 98 studies, providing a broad and diverse sample to assess the research inefficiencies related to the Framingham Wilson CHD risk rule.

• Use of meta-analyses: The authors conducted meta-analyses to synthesize data from multiple studies, providing more precise estimates of the risk rule's performance across different populations and settings. This approach adds to the strength of the study, as it enables a more comprehensive understanding of the risk rule's overall performance.

• Assessment of study characteristics and reporting: The paper thoroughly evaluates the characteristics of the included studies, as well as their reporting of performance measures. This assessment helps identify areas where research practices could be improved to address the observed inefficiencies.

8. However, there was Lack of exploration of reasons behind inefficiencies: The paper does not delve deeply into the underlying reasons for the observed research inefficiencies. Understanding these factors could help inform more specific recommendations for improving research practices in this area. The authors could conduct further analyses or interviews with researchers to better understand the factors contributing to the observed research inefficiencies. This could help inform targeted recommendations for improving research practices.

We appreciate and agree with the reviewer’s comment. In fact, for the derivation step, we published a study exploring one of the key inefficiencies and evaluated why researchers keep deriving new cardiovascular CPRs despite many similar CPRs already exist (2). However, exploring reasons why the research inefficiencies occur in external validation studies and identifying factors associated with these inefficiencies, which could be used to develop targeted interventions, are beyond the aim and scope of our study. Therefore, we revised the research and clinical implications section to discuss this. 

• Line 999-1004: “In this study, we identified two key inefficiencies in the external validation sept of cardiovascular CPR development. Future studies (e.g. qualitative interviews or analysis of CPR registry) could further explore causes and mechanisms for these inefficiencies. Understanding factors that influence these inefficiencies to occur might help develop targeted ways to prevent them and increase the chance for external validation studies to fully contribute to knowledge and practice.”

The writing quality of the paper is generally good, with clear language and appropriate terminology. The organization of the paper follows a logical structure, with a clear introduction, methods, results, and discussion sections. The clinical usefulness of the paper is evident, as it informs researchers and clinicians about the potential limitations of the risk rule and the need for updating it.

In conclusion, the paper provides a valuable contribution to the literature by highlighting the research inefficiencies in studies evaluating the Framingham Wilson CHD risk rule. This could potentially lead to improvements in future research and help enhance the overall quality and utility of clinical prediction rules in the field of cardiovascular disease.

References

1. Steyerberg EW. Clinical prediction models : a practical approach to development, validation, and updating. New York: Springer; 2009. xxviii, 497 p. p.

2. Ban JW, Wallace E, Stevens R, Perera R. Why do authors derive new cardiovascular clinical prediction rules in the presence of existing rules? A mixed methods study. PloS one. 2017;12(6):e0179102.

---

## [Decision Letter · Decision Letter 2]

31 Jul 2023

PONE-D-22-18030R2Research inefficiencies in external validation studies of the Framingham Wilson coronary heart disease risk rule: A systematic reviewPLOS ONE

Dear Dr. Ban,

Thank you for submitting your manuscript to PLOS ONE. After careful consideration, we feel that it has merit but does not fully meet PLOS ONE’s publication criteria as it currently stands. Therefore, we invite you to submit a revised version of the manuscript that addresses the points raised during the review process.

I took over as Academic Editor after the 2nd revision was submitted. I do not want to interfere in the review process carried out up to now, as the reviewers are obviously happy with the revised manuscript (besides one minor point of Reviewer 4 that should be addressed).

However, during reading the paper, I found many inaccuracies. This leads to the question, whether we can really trust those results of your systematic review that we cannot simply check, e.g. by counting. Please consider, that you have already submitted three versions of your manuscript. I wonder that after reading and approving the manuscript before submission, how can such inaccuracies still be present in the third version?

We look forward to receiving your revised manuscript.

Kind regards,

Harald Heinzl

Academic Editor

PLOS ONE

Editor Comments:

lines 141-144 and 216-217: How does your third study selection criterion match with your CHD definitions for meta-analysis (Table S1)?

lines 216-217: Provide some solid arguments why a meta-analysis makes sense when the involved studies differ so much in their outcome definitions (Table S1).

line 250, Table 1: Describe the difference between a "Risk equation" and a "Risk score".

line 252: "Journal" instead of "Jounral"

line 261: "16 of 40 (40 %)", in Table 2 we find "17 (42.5)"

line 270 and Fig. 2: "27 studies", in line 282 we find "26 studies". Is this a simple typo? Or did you omit one study in the post hoc citation analysis (which one and why)?

line 327: "Six studies", in Table 3 we can find 8 studies.

line 339: "ten (43.5%)", in Table 3 we can only find 9 studies.

line 340: "13 (56.5%)", in Table 3 we find 14 studies.

lines 427-434: In the manuscript, you criticize the studies that they don't follow the TRIPOD. Now you explain that all except three studies predate TRIPOD as the statement appeared in 2015. This explanation should appear much earlier in the paper (e.g. in the Indroduction section).

line 431: "from" instead of "form"?

line 433: "understanding of" instead of "understandin"?

line 444: "step" instead of "sept"?

lines 478-480: This sentence is hard to understand and should be reformulated.

lines 760-761: Supporting information S5 and S6 must be appropriately mentioned in the manuscript.

Fig. 1: "measure" instead of "meausre"

Fig. 3-A: Shouldn't it read "Orford (...)" instead of "outcome"?

Fig. 3-B: "Belgium" instead of "Belbium"

Figures 3-A and 3-B: Using the Fowkes (2008) paper, I tried to confirm the 777 observed participants with "outcome" of the ARIC study by way of example, and I failed. How have the obsered AND predicted participants been computed in the manuscript for the studies of Figures 3-A and 3-B? You could provide this information as supporting information S7.

Figures 3-A and 3-B: Which formula did you use for the confidence interval of the P/O ratio? You could provide this information as supporting information S7.

Figures 3-A, 3-B and 4: According to lines 304 and 309, N is the total number of participants. However, when checking the Orford ("outcome") study which is the first study of both Figures 3-A and 4, there are 5,611 versus 1,393 participants, respectively. Is there a typo somewhere or is the former number the person-years-under-risk? A thorough check of all the reported participant numbers is necessary.

Fig. 4: According to Table 3, there should be 16 "bold" studies, not just 9. It seems that Western and Southern Europe have been forgotten.

Fig. 4, USA: Two studies with more than 14,000 participants have rather wide confidence intervals compared to studies with much smaller number of participants. Is there a sound formal explanation for this fact?

Fig. 4: How has the c statistic been computed? Which formula did you use for the confidence interval of the c statistic? You could provide this information as supporting information S7.

Reviewers' comments:

Reviewer's Responses to Questions

**Comments to the Author**

1. If the authors have adequately addressed your comments raised in a previous round of review and you feel that this manuscript is now acceptable for publication, you may indicate that here to bypass the “Comments to the Author” section, enter your conflict of interest statement in the “Confidential to Editor” section, and submit your "Accept" recommendation.

Reviewer #1: All comments have been addressed

Reviewer #4: (No Response)

2. Is the manuscript technically sound, and do the data support the conclusions?

Reviewer #1: Yes

Reviewer #4: Yes

3. Has the statistical analysis been performed appropriately and rigorously? 

Reviewer #1: Yes

Reviewer #4: I Don't Know

4. Have the authors made all data underlying the findings in their manuscript fully available?

Reviewer #1: Yes

Reviewer #4: Yes

5. Is the manuscript presented in an intelligible fashion and written in standard English?

Reviewer #1: Yes

Reviewer #4: Yes

6. Review Comments to the Author

Reviewer #1: Thank you for your responses.

...............................................................................

Reviewer #4: I am generally satisfied with the responses but feel the authors could still include more information as to why the regional areas (e.g., US and Europe) were selected. They have provided an excellent response to me, the reviewer, but this is not included in the paper, which would be helpful to the reader.

I am unable to comment on the statistical aspects of the paper.

I found the other reviewers’ comments (and author responses) very insightful. I echo my earlier comment that this is a very practical and useful article.

7. PLOS authors have the option to publish the peer review history of their article (what does this mean?). If published, this will include your full peer review and any attached files.

Reviewer #1: No

Reviewer #4: No

---

## [Author Response · Author response to Decision Letter 2]

27 Oct 2023

Academic Editor’s Comments:

1. lines 141-144 and 216-217: How does your third study selection criterion match with your CHD definitions for meta-analysis (Table S1)?

The Framingham Heart Study defined the coronary heart disease (CHD) outcome and its components using explicit and detailed criteria [1], which were then used when deriving the Framingham Wilson CHD risk rule [2]. It would have been simple if studies externally validating the Framingham Wilson CHD risk rule used the same definitions and criteria. However, no external validation study exactly followed the ones from the Framingham Heart Study. 

We suspect that heterogeneous outcome definitions were used because, since the Framingham Heart Study defined these outcomes several decades ago, standard practices for evaluating patients for CHD have evolved greatly [3-5]. For example, the Framingham Heart Study's definition of acute myocardial infarction was based on serial changes in electrocardiogram findings and elevated serum markers such as glutamic oxaloacetic transaminase (GGT) and lactic dehydrogenase (LDH) [1]. However, criteria such as GGT and LDH have become obsolete in clinical practice and were replaced by better diagnostic tools such as troponin-I assay, exercise stress test, and scintigraphy, which were often used to determine CHD outcomes in external validation studies. 

Further, we consider it is useful to study whether the Framingham Wilson CHD risk rule can accurately predict CHD in changing times, locations, and circumstances (please also refer to our response to comment #2 for more details). Therefore, we included studies that evaluated either CHD or hard CHD, as described in our third selection criteria. For example, a study by Herrera et al. [6] defined angina pectoris as "compatible chest pain history plus a positive exercise stress test, scintigraphy, or coronary arteriogram." Although different from the Framingham Heart Study's original definition, we determined that Herrera et al.'s outcome definition of angina pectoris [6] was sensible because it was consistent with modern standards for evaluating stable patients with chest pain [3-5]. We added the following sentence to the method section to explain to readers. 

• Line 155-158: “When studies defined CHD outcomes using criteria different from the ones used by the Framingham Heart Study (40) due to evolving practice standards over time, we judged whether the modifications were relevant to current clinical practice.” 

2. lines 216-217: Provide some solid arguments why a meta-analysis makes sense when the involved studies differ so much in their outcome definitions (Table S1).

Internal validation assesses the reproducibility of a clinical prediction rule (CPR) in the same population where the CPR is derived [7, 8]. On the other hand, external validation studies evaluate how well the CPR performs when tested in populations different from but "plausibly related to" the derivation population [8, 9]. In practice, it means that, by conducting external validation studies, CPRs are applied to circumstances or settings of the CPRs' intended use that might differ regarding locations, time, methods, or disease spectrum [8, 10]. Therefore, heterogeneity is inherent and expected in external validation studies, which might include one that originates from differences in predictor and outcome definitions [10]. 

Similarly, meta-analysis results of external validation studies summarise the average performance of the CPR across various circumstances or settings and often contain heterogeneity due to the differences in predictor and outcome measurements [11, 12]. Therefore, having heterogeneity is not necessarily a contraindication for conducting meta-analyses of external validation studies but is rather an opportunity to explore potential sources of variation [12], as we did in our study for geographic locations. Although a similar subgroup analysis (or meta-regression) investigating the heterogeneity due to different outcome definitions would have been possible, this was not one of our aims. Furthermore, many external validation studies in our study were also included in the existing meta-analyses of the Framingham Wilson CHF risk rule [13, 14]. 

3. line 250, Table 1: Describe the difference between a "Risk equation" and a "Risk score".

Wilson et al. [2] presented their CHD risk rule in two formats: (1) regression equations that estimate the 10-year risk of developing CHD using coefficients of predictor variables and (2) simplified scoring systems that categorise the 10-year risk of developing CHD based on the sum of points assigned to applicable predictor variables. We added the following texts in the legend of Table 1. 

Line 271-272: “a Risk equation: regression equations that estimate the 10-year risk of developing CHD using coefficients of predictor variables.” 

“b Risk score: simplified scoring systems that categorise the 10-year risk of developing CHD based on the sum of points assigned to applicable predictor variables.”

4. line 252: "Journal" instead of "Jounral"

Apologies for the error. We corrected the misspelling in the legend of Table 1. 

5. line 261: "16 of 40 (40 %)", in Table 2 we find "17 (42.5)"

Apologies for the error. In the previous revision, when the contents of Table 2 were revised with an additional study found in the updated search, we missed reflecting it into the text. We revised the text and provided the correct data. 

• Line 281-282: “… c statistic was reported with a 95% confidence interval only in 17 of 40 (42.5%) external validation studies, …”

6. line 270 and Fig. 2: "27 studies", in line 282 we find "26 studies". Is this a simple typo? Or did you omit one study in the post hoc citation analysis (which one and why)?

Apologies for the error. When the texts were updated in the previous revision, we missed updating the figure. Therefore, the values in Fig. 2 have now been updated.

7. line 327: "Six studies", in Table 3 we can find 8 studies.

Six external validation studies could not be included in the meta-analysis of the P/O ratio because they neither reported a P/O ratio nor provided sufficient information about the data to estimate it. These six studies (Mainous 2007 (51), Gander 2014 (54), Van Der Heijden 2009 (14), Merry 2012 (59), Guckelberger 2006 (62), and Protopsaltis 2004 (67)) are categorised in the table as “Not available*” in the fourth column.

8. line 339: "ten (43.5%)", in Table 3 we can only find 9 studies.

These ten studies are Orford 2002 (50), Rodondi 2012 (48), Emapana 2003 (55), Simmons 2008 (56), Brunner 2010 (57), Empana 2003 (60), Becker 2008 (63), Marrugat 2007 (24), Cañón-Barroso 2007 (65), Calvo-Hueros (66). For Simmons 2008 (56), conducted with 10.295 participants from the EPIC-Norfolk cohort, the study was eligible for the meta-analysis of c statistic. However, it was ineligible for the meta-analysis of the P/O ratio because Sivapalaratnam (2010 Oct), using a much larger sample of 22,841 participants from the same cohort, was included. To avoid misunderstanding and make the intent clear, we revised the sentence.

• Line 363-365: “…of 23 studies that aimed to externally validate the Framingham Wilson CHD risk rule, only ten (43.5%) contributed to understanding its performance in eligible meta-analyses of P/O ratio and c statistic.”

9. line 340: "13 (56.5%)", in Table 3 we find 14 studies.

Similar to the comment 8, we consider Simmons 2008 (56) fully contributed to its eligible meta-analysis because it was only eligible for the meta-analysis of c statistic. Therefore, we believe 13 of 23 (56.5%) did not contribute to eligible meta-analyses to their full potential. 

10. lines 427-434: In the manuscript, you criticize the studies that they don't follow the TRIPOD. Now you explain that all except three studies predate TRIPOD as the statement appeared in 2015. This explanation should appear much earlier in the paper (e.g. in the Indroduction section).

As recommended, we acknowledged in the introduction section that the TRIPOD statement was published recently and the authors of studies predating it did not have reporting guidance.

• Line 94-100: “According to the TRIPOD statement published in 2015, calibration and discrimination measures "should be reported in all prediction model papers" (29). However, systematic reviews have shown that external validation studies often do not report recommended performance measures, especially measures of calibration (8, 30, 31). Most studies included in these systematic reviews predate the TRIPOD statement, and authors might not have guidance for appropriate reporting. However, it is still true that when external validation studies do not report relevant performance measures …”

11. line 431: "from" instead of "form"?

Apologies. We corrected the misspelling. 

• Line 456-458: “… our findings should not be interpreted as an assessment of reporting quality against the recommendations from the TRIPOD statement.”

12. line 433: "understanding of" instead of "understandin"?

Apologies. We corrected the misspelling. 

• Line 459-460: “… so that their results could contribute to the understanding the overall performance of the Framingham Wilson CHD risk rule.”

13. line 444: "step" instead of "sept"?

Apologies. We corrected the misspelling. 

• Line 471-472: “…we identified two key inefficiencies in the external validation step of cardiovascular CPR development.”

14. lines 478-480: This sentence is hard to understand and should be reformulated.

We revised the conclusion section to make the message clear. 

• Line 502-514: “The focus of cardiovascular CPR research should shift from deriving many new CPRs to externally validating existing CPRs and updating them when needed because the potential end-users, such as patients, clinicians, researchers, and decision-makers, need to understand the generalizability of the CPRs. However, we have shown that conducting more external validation studies of a CPR does not always strengthen the CPR's evidence of generalizability. Further, we demonstrated that dismissing a CPR when it performs poorly can lead to the creation of new, potentially redundant CPRs. Therefore, in addition to shifting the focus of CPR research, the authors of external validation studies should ensure that they are using appropriate design and methods and providing sufficient information about the conducts and results so that the evidence they generate can help the end-users about the generalizability of the CPRs.”

15. lines 760-761: Supporting information S5 and S6 must be appropriately mentioned in the manuscript.

We added the following sentences to the methods section to introduce supporting information: S5 and S6. Please note that supporting documents are re-ordered to match the order in which they appear in the main text. 

• Line 247-249: “We prepared this report following all applicable recommendations from the Preferred Reporting Items for Systematic Reviews and Meta-Analyses (PRISMA) statement (48) as summarised in S3 Appendix. Data from our study are provided in S4 Dataset.”

16. Fig. 1: "measure" instead of "meausre" 

We corrected the misspelling in Fig. 1 to read “measure.” 

17. Fig. 3-A: Shouldn't it read "Orford (...)" instead of "outcome"?

We corrected the misspelling in Fig. 3-A to read “Orford (2002 Jul).” 

18. Fig. 3-B: "Belgium" instead of "Belbium"

We corrected the misspelling in Fig. 3-B to read “Belgium.”

19. Figures 3-A and 3-B: Using the Fowkes (2008) paper, I tried to confirm the 777 observed participants with "outcome" of the ARIC study by way of example, and I failed. How have the observed AND predicted participants been computed in the manuscript for the studies of Figures 3-A and 3-B? You could provide this information as supporting information S7.

The Framingham Wilson CHD risk rule predicts a 10-year risk of CHD outcome. However, some studies assessed the CHD outcome with follow-up durations different from 10 years. For example, the ARIC cohort from Fowkes (2008) reported 571 CHD events with a median follow-up of 13.1 years for men and 362 CHD events with a median follow-up of 13.2 years for women. We added S2 Appendix to explain how these data were used to estimate CHD events anticipated with a 10-year follow-up.

20. Figures 3-A and 3-B: Which formula did you use for the confidence interval of the P/O ratio? You could provide this information as supporting information S7.

We added S2 Appendix to explain how we conducted meta-analyses using methods described by Debray et al. [15].

21. Figures 3-A, 3-B and 4: According to lines 304 and 309, N is the total number of participants. However, when checking the Orford ("outcome") study which is the first study of both Figures 3-A and 4, there are 5,611 versus 1,393 participants, respectively. Is there a typo somewhere or is the former number the person-years-under-risk? A thorough check of all the reported participant numbers is necessary.

Apologies for these errors. We have checked all the data, calculations, and results presentation in these figures and made corrections if needed. 

Previous version of Fig 3-A

Revised Fig 3-A

Previous version of Fig 3-B

Revised Fig 3-B

Previous version of Fig 4

Revised Fig 4

22. Fig. 4: According to Table 3, there should be 16 "bold" studies, not just 9. It seems that Western and Southern Europe have been forgotten.

We revised Fig. 4 by changing the characters of five relevant Western and Southern European studies to make the information consistent with Table 3. There are now 16 “bold” studies in Fig. 4.

23. Fig. 4, USA: Two studies with more than 14,000 participants have rather wide confidence intervals compared to studies with much smaller number of participants. Is there a sound formal explanation for this fact?

Thank you for pointing these out. We reviewed all calculations related to the meta-analysis of c statistic in Fig. 4 and made appropriate changes. 

For Dene 2007, the number of previously reported participants was incorrect. Although the study included a total of 14,749 participants, only 1,264 participants were used in the calculation. Therefore, we corrected the number of participants for Dene 2007. 

On the other hand, for Mainous 2007, the reported number of participants presented was correct. However, an incorrect value was used in our meta-analysis by mistake. Now, the appropriate confidence interval for the c statistic is presented.

24. Fig. 4: How has the c statistic been computed? Which formula did you use for the confidence interval of the c statistic? You could provide this information as supporting information S7.

We added S2 Appendix to explain how we conducted meta-analyses using methods described by Debray et al. [15].

Reviewer #4: I am generally satisfied with the responses but feel the authors could still include more information as to why the regional areas (e.g., US and Europe) were selected. They have provided an excellent response to me, the reviewer, but this is not included in the paper, which would be helpful to the reader.

As advised by the reviewer, we modified our response to the reviewer’s comment from the previous revision and included the following texts in the paper to help the readers understand why studies were meta-analysed by regional areas.

Line -: “Arguably, a meta-analysis of external validation studies should include studies conducted in populations that are “plausibly related” (45), which is not always easy to determine clearly. In our study, we took a conservative approach and included in meta-analyses only the studies conducted in the USA and European geographic regions with comparable cardiovascular disease burden, the prevalence of risk factors, and healthcare environment (46): the UK, Northern Europe, Western Europe, and Southern Europe. We meta-analysed the Predicted/Observed (P/O) event ratio …”

1. Kannel WB, Wolf PA, Garrison RJ, Cupples LA, D'Agostino RB, National Heart L, et al. The Framingham study : an epidemiological investigation of cardiovascular disease / Section 34 : Some risk factors related to the annual 

---

## [Editor Report · Decision Letter 3]

20 Nov 2023

PONE-D-22-18030R3Research inefficiencies in external validation studies of the Framingham Wilson coronary heart disease risk rule: A systematic reviewPLOS ONE

Dear Dr. Ban,

Thank you for submitting your manuscript to PLOS ONE. After careful consideration, we feel that it has merit but does not fully meet PLOS ONE’s publication criteria as it currently stands. Therefore, we invite you to submit a revised version of the manuscript that addresses the points raised during the review process.

Thank you for your revision. Unfortunately, you did not address a major issue properly.

Looking at your Figures 3A, 3B and 4, there is an enormous heterogeneity of the study results. In particular, 3A (USA) and 3B (Western Europe) show extremely contradicting study results ("qualitative" heterogeneity). Therefore I asked:

Provide some solid arguments why a meta-analysis makes sense when the involved studies differ so much in their outcome definitions? You provided an interesting answer for me but not for your audience (future readers of your paper).

That is, you should address the question in the paper, thereby referencing to Figures 3A, 3B and 4, in particular 3A (USA) and 3B (Western Europe). Provided that this enormous heterogeneity is not due to different outcome definitions, some additional explanation is needed in any case.

Minor point: You state that you have corrected the misspelling in Fig.1 (measure instead of meausre). I suggest that you check again.

Remark: Regarding the Simmons 2008 (56) and the Sivapalaratnam (2010 Oct) study (Table 3), I want to mention that the readers of the paper will not be able to guess the details.

We look forward to receiving your revised manuscript.

Kind regards,

Harald Heinzl

Academic Editor

PLOS ONE

---

## [Author Response · Author response to Decision Letter 3]

7 Feb 2024

1. Thank you for your revision. Unfortunately, you did not address a major issue properly. Looking at your Figures 3A, 3B and 4, there is an enormous heterogeneity of the study results. In particular, 3A (USA) and 3B (Western Europe) show extremely contradicting study results (“qualitative” heterogeneity). Therefore I asked: Provide some solid arguments why a meta-analysis makes sense when the involved studies differ so much in their outcome definitions? You provided an interesting answer for me but not for your audience (future readers of your paper). That is, you should address the question in the paper, thereby referencing to Figures 3A, 3B and 4, in particular 3A (USA) and 3B (Western Europe). Provided that this enormous heterogeneity is not due to different outcome definitions, some additional explanation is needed in any case.

We accept that the summary statistics are of unclear value when heterogeneity is high. Since the summary statistics are largely incidental to our paper and not relevant to either our primary or secondary research question, we have removed them from the Forest plots.

We have therefore added the following text to the Methods section:

• Line 227-230: “At the suggestion of a reviewer, we display results in forest plots without summary estimates when heterogeneity is high, since summary estimates from meta-analyses are incidental to our study aims rather than directly relevant to our research questions.”

In addition, we have replaced the follow paragraph in the Results section,

“The summary P/O ratio from the meta-analysis of 17 studies conducted in the USA was 1.065 with a 95% confidence interval of 0.763 - 1.486, as presented in Fig 3-A. The summary P/O ratios (95% confidence intervals) from the meta-analyses of studies conducted in Europe were 1.632 (1.346 - 1.978) for the UK, 1.846 (1.246 - 2.737) for Northern Europe, 1.399 (0.920 - 2.128) for Western Europe, and 2.186 (1.868 - 2.555) for Southern Europe, as illustrated in Fig 3-B.”

with

• Line 310-311: “As presented in Figs 3-A and 3-B, the P/O ratios that were obtained showed high heterogeneity, with an I2 statistic greater than 80% in all analyses.”

Similarly, we have replaced

“The meta-analysis results of the c statistic of the Framingham Wilson CHD risk rule are presented in Fig 4. The summary c statistic and 95% confidence interval from the meta-analysis of seven studies conducted in the USA were 0.695 and 0.643 - 0.742. The summary c statistic (95% confidence intervals) from the meta-analyses of studies conducted in Europe were 0.694 (0.667 - 0.720) for the UK, 0.703 (0.621-0.773) for Northern Europe, 0.691 (0.663 - 0.718) for Western Europe, and 0.658 (0.603 - 0.709) for Southern Europe.” 

with 

• Line 333-340: “Forest plots of the c statistic of the Framingham Wilson CHD risk rule are presented in Fig 4. Studies conducted in the USA, Northern Europe, and Southern Europe had high heterogeneity with a corresponding I2 statistic of 96.98%, 96.50%, and 79.06%, respectively. The summary c statistic (95% confidence interval) from the meta-analysis of three studies conducted in the UK was 0.699 (0.680 - 0.718) with an I2 statistic of 30.33%. The summary c statistic (95% confidence intervals) from the meta-analyses of five Western European studies was 0.692 (0.660 - 0.722) with an I2 statistic of 49.00%.”

Further, we have added the following text to the Summary of Findings subsection of the Discussion:

• Line 406-415: “The heterogeneity between studies suggests that, even when many previous validations have been conducted in a particular geographic location, there is still potential to learn something new due to differences in populations, changes in predictor and outcome definitions, and temporal evolution of clinical practice (71). In particular, since the Framingham Heart Study defined these outcomes several decades ago, standard practices for diagnosing CHD have evolved greatly (72-74). Therefore, we do not argue that the validation studies were wasteful in themselves. Rather, we encourage authors of validation studies to maximise the value of their study: by publishing comprehensive validation statistics and by publishing an updated (“recalibrated”) rule when indicated.”

and the following text to the Strengths and Limitations subsection:

• Line 485-488: “We only considered overall calibration (P/O ratio) and discrimination (c statistic) in particular; good validation should consider three or more dimensions (89, 90). However, focussing on two commonly reported aspects of validation did not prevent us from demonstrating research inefficiencies in this literature. Lastly, some…”

2. Minor point: You state that you have corrected the misspelling in Fig.1 (measure instead of meausre). I suggest that you check again.

We corrected all misspells in Fig 1. 

3. Remark: Regarding the Simmons 2008 (56) and the Sivapalaratnam (2010 Oct) study (Table 3), I want to mention that the readers of the paper will not be able to guess the details.

We added the following sentences to the main text and revised the legend of Table 3 to explain two studies, Vaidya 2007 (53) and Simmons 2008 (54), were ineligible for the meta-analysis of the P/O ratio. We also recognise that P/O ratios can potentially be obtained using methods other than those used in our study. Therefore, we discussed it in the Strengths and Limitations section.

• Line 358-361: “All 23 studies were eligible to be included in the meta-analysis of c static. However, only 21 of 23 studies were eligible for the meta-analysis of the P/O ratio; two studies, Vaidya 2007 (53) and Simmons 2008 (54), were ineligible because they were conducted using a subset of data from the same cohort used by another eligible study.”

• Line 488-494: “Also, we followed the approaches of Debray et al (47), as described in S2 Appendix, for extracting P/O ratios; arguably, a systematic reviewer with sufficient software and mathematical knowledge could also estimate P/O ratios from published calibration plots, albeit inexactly; this would increase the number of external validation studies of the Framingham Wilson CHD risk rule that contributed to understanding the performance in eligible meta-analyses of P/O ratio and c statistic from 10 of 23 (43.5%) to 11 (47.8%), but not change our overall conclusions.”

---

## [Decision Letter · Decision Letter 4]

11 Jun 2024

PONE-D-22-18030R4Research inefficiencies in external validation studies of the Framingham Wilson coronary heart disease risk rule: A systematic reviewPLOS ONE

Dear Dr. Ban,

Thank you for submitting your manuscript to PLOS ONE. After careful consideration, we feel that it has merit but does not fully meet PLOS ONE’s publication criteria as it currently stands. Therefore, we invite you to submit a revised version of the manuscript that addresses the points raised during the review process.  

We look forward to receiving your revised manuscript.

Kind regards,

Daniel Antwi-Amoabeng

Academic Editor

PLOS ONE

Journal Requirements:

Additional Editor Comments:

Please complete revision as suggested by Reviewer 6:

1. Figures 3A and 3B are difficult to understand. Providing a detailed legend would help the reader interpret the figures more easily. Additionally, the x and y axis labels are missing. The authors could also better address the large heterogeneity in the dataset.

2) The current language of the article is too harsh and requires major revisions for language and content.

Thank you.

Reviewers' comments:

Reviewer's Responses to Questions

**Comments to the Author**

1. If the authors have adequately addressed your comments raised in a previous round of review and you feel that this manuscript is now acceptable for publication, you may indicate that here to bypass the “Comments to the Author” section, enter your conflict of interest statement in the “Confidential to Editor” section, and submit your "Accept" recommendation.

Reviewer #4: All comments have been addressed

Reviewer #6: All comments have been addressed

2. Is the manuscript technically sound, and do the data support the conclusions?

Reviewer #4: (No Response)

Reviewer #6: Partly

3. Has the statistical analysis been performed appropriately and rigorously? 

Reviewer #4: (No Response)

Reviewer #6: Yes

4. Have the authors made all data underlying the findings in their manuscript fully available?

Reviewer #4: (No Response)

Reviewer #6: Yes

5. Is the manuscript presented in an intelligible fashion and written in standard English?

Reviewer #4: (No Response)

Reviewer #6: No

6. Review Comments to the Author

Reviewer #4: (No Response)

Reviewer #6: In this study, the authors attempted to highlight whether external validation studies of the Framingham Wilson coronary heart disease (CHD) risk rule contributed to generating evidence to their full potential. The general concept and methodology of this study are intriguing, and further studies based on these findings may contribute to the Framingham Wilson CHD risk rule's generalizability to their full potential. However, I have some queries about the study. I feel the authors need to carefully mention them in the manuscript.

1. Figures 3A and 3B are difficult to understand. Providing a detailed legend would help the reader interpret the figures more easily. Additionally, the x and y axis labels are missing. The authors could also better address the large heterogeneity in the dataset.

2) The current language of the article is too harsh and requires major revisions for language and content.

7. PLOS authors have the option to publish the peer review history of their article (what does this mean?). If published, this will include your full peer review and any attached files.

Reviewer #4: No

Reviewer #6: No

---

## [Author Response · Author response to Decision Letter 4]

21 Aug 2024

PONE-D-22-18030R4

Research inefficiencies in external validation studies of the Framingham Wilson coronary heart disease risk rule: A systematic review

Additional Editor Comments:

Please complete revision as suggested by Reviewer 6:

1. Figures 3A and 3B are difficult to understand. Providing a detailed legend would help the reader interpret the figures more easily. Additionally, the x and y axis labels are missing. 

Response: 

We appreciate the reviewer’s comment. Accordingly, we revised the legend and explained the components of Figures 3A and 3B. Also, we added the x-axis label to indicate the figures presented the Predicted/Observed (P/O) ratio. We removed numbering on the y-axis to make it clear that each P/O ratio in forest plots corresponds to the study presented in the same row. For consistency, we made similar changes to Fig 4 and its legend. 

Line 332–347 : “Study: the last name of the study's first author (publication year and month). Studies with an explicit aim to externally validate the Framingham Wilson coronary heart disease rule are presented in bold characters; Country: the country where the study was conducted; Cohort: the specific cohort where the Framingham Wilson coronary heart disease rule was applied; Predicted, n/N: the number of participants with the predicted outcome (n) and the total number of participants (N); Observed, n/N: the number of participants with the observed outcome (n) and the total number of participants (N).”

2. The authors could also better address the large heterogeneity in the dataset.

Regarding the large heterogeneity noted in the results of the included studies, we revised the manuscript to make the following points clear to the readers: (1) heterogeneity, sometimes to a large degree, is often assumed to be present in external validation studies of clinical prediction rules (CPRs) and (2) we did not aim to summarise the performance of the Framingham Wilson CHD risk rule (and meta-analyses were not directly relevant to our research questions), we did not explore the heterogeneity extensively. 

Line 484-486:“Heterogeneity is often expected among external validation studies of clinical prediction rules. We did not explore the source of heterogeneity in our data because evaluating the predictive performance of the Framingham Wilson CHD risk rule was not one of our aims.”

3. The current language of the article is too harsh and requires major revisions for language and content.

We would like to emphasize that it was not our intention to use harsh language, nor was it to be disrespectful to researchers externally validating clinical prediction rules. On the contrary, we admire and appreciate their work, which provides essential evidence that is a prerequisite for implementing high-quality clinical prediction rules. Our hope was to point out areas where making certain changes could enhance the value of their work. Therefore, we tried our best to identify and revise the parts of the manuscript that might have the potential to be unintentionally interpreted as too harsh criticism.

Abstract 

(line 43): changed “Most… failed to” to “Many… did not”.

Introduction 

(line 71): changed “wasted” to “lost”; changed “promote the proliferation” to “inadvertently contribute to creating”

(line 103-104) changed “According to the TRIPOD statement ... should be reported in all” to

“The TRIPOD statement ... recommends that authors report”

Discussion 

(line 422): changed ‘fail to’ to ‘could not’

(line 427) changed ‘research waste and inefficiencies’ to ‘research inefficiencies’

(line 435) inserted ‘they still created valuable knowledge’

---

## [Editor Report · Decision Letter 5]

29 Aug 2024

Research inefficiencies in external validation studies of the Framingham Wilson coronary heart disease risk rule: A systematic review

PONE-D-22-18030R5

Dear Dr. Ban,

We’re pleased to inform you that your manuscript has been judged scientifically suitable for publication and will be formally accepted for publication once it meets all outstanding technical requirements.

Kind regards,

Daniel Antwi-Amoabeng, MD, MSc

Academic Editor

PLOS ONE

Additional Editor Comments (optional):

Thank you for carefully addressing the issues raised by the various reviewers.
---

## [Editor Report · Acceptance letter]

5 Sep 2024

PONE-D-22-18030R5 

PLOS ONE

Dear Dr. Ban, 

I'm pleased to inform you that your manuscript has been deemed suitable for publication in PLOS ONE. Congratulations! Your manuscript is now being handed over to our production team.

Kind regards, 

on behalf of

Dr. Daniel Antwi-Amoabeng 

Academic Editor

PLOS ONE